

# A systematic literature review of hate speech identification on Arabic Twitter data: research challenges and future directions

Ali Alhazmi[1,2], Rohana Mahmud[1], Norisma Idris[1], Mohamed Elhag Mohamed Abo[3] and Christopher Eke[4]

[1] Faculty of Computer Science and Information Technology, Universiti Malaya, Kuala Lumpur, Malaysia
[2] Department of Information Technology and Security, Jazan University, Jazan, Saudi Arabia
[3] Department of Computer Science, The Future University, Khartoum, Sudan
[4] Department of Computer Science, Faculty of Computing, Federal University of Lafia, Lafia, Nasarawa State, Nigeria

Corresponding authors
Rohana Mahmud,
rohanamahmud@um.edu.my
Norisma Idris, norisma@um.edu.my

## ABSTRACT

The automatic speech identification in Arabic tweets has generated substantial attention among academics in the fields of text mining and natural language processing (NLP). The quantity of studies done on this subject has experienced significant growth. This study aims to provide an overview of this field by conducting a systematic review of literature that focuses on automatic hate speech identification, particularly in the Arabic language. The goal is to examine the research trends in Arabic hate speech identification and offer guidance to researchers by highlighting the most significant studies published between 2018 and 2023. This systematic study addresses five specific research questions concerning the types of the Arabic language used, hate speech categories, classification techniques, feature engineering techniques, performance metrics, validation methods, existing challenges faced by researchers, and potential future research directions. Through a comprehensive search across nine academic databases, 24 studies that met the predefined inclusion criteria and quality assessment were identified. The review findings revealed the existence of many Arabic linguistic varieties used in hate speech on Twitter, with modern standard Arabic (MSA) being the most prominent. In identification techniques, machine learning categories are the most used technique for Arabic hate speech identification. The result also shows different feature engineering techniques used and indicates that N-gram and CBOW are the most used techniques. F1-score, precision, recall, and accuracy were also identified as the most used performance metric. The review also shows that the most used validation method is the train/test split method. Therefore, the findings of this study can serve as valuable guidance for researchers in enhancing the efficacy of their models in future investigations. Besides, algorithm development, policy rule regulation, community management, and legal and ethical consideration are other real-world applications that can be reaped from this research.

## INTRODUCTION

Over the past 10 years, Twitter and other social media have experienced exponential growth. According to *Zhang & Luo (2019)*, these social media sites encourage user privacy and enable them to freely express themselves, which fosters the development and dissemination of hate speech (HS). With 300 million participants each month, Twitter is one of the most widely used social networking sites. HS is becoming an increasing phenomenon on social media. According to *Silva et al. (2016)*, HS is routinely disseminated on Twitter despite being widespread and pertinent. As a source of information for studies on obscene language, it is currently one of the most widely used social networks to automatically detect HS in textual data (*Founta et al., 2019*; *Magu, Joshi & Luo, 2017*; *Mondal et al., 2018*). Users become hostile as a result, which leads to serious confrontations in the real world and has an impact on enterprises. Social media platforms frequently remove offensive posts, stopping their publication. Users can effectively express their feelings using the language that they speak fluently. Besides the English language, Arabic is regarded as one of the official languages in 22 nations including African, Gulf, and Middle Eastern regions. Arabic ranks as the fifth most extensively used language in the globe with over 422 million native and non-native speakers (*Al-Anzi & AbuZeina, 2022*; *Elnagar et al., 2021*). Arabic-language messages from Twitter are the focus of this study because it is a well-known and recognized language globally and the most accessible data source (*Arango, Pérez & Poblete, 2019*). The richness and complexity of the Arabic language at the morphological, lexical, and orthographic levels, as well as its unique characteristics, are, in our opinion (*Darwish, Magdy & Mourad, 2012*), provide some particular difficulties that may make it more difficult to identify hate speech. The problem is further complicated by the wide range of Arabs who use social media and speak dialectic Arabic. In truth, Arabic dialects are numerous, differing not just inside a single country but also between countries. As a result, various words with similar spellings have diverse meanings in various dialects and geographical areas.

Consequently, automating the process of identifying hate speech online is necessary because manual filtering is rigid. The response time is directly impacted by non-automated processes, but a computer-based method can complete this work more quickly than humans can. Consequently, supporting automatic textual hate speech identification technology is therefore crucial. These findings have spurred natural language processing (NLP) research.

The amount of research on hate speech is growing rapidly in the literature. *Schmidt & Wiegand (2017)* asserted that supervised document categorization using NLP and machine learning has been used to classify this issue. Based on a review of several definitions presented in the literature on this subject, reference (*Langham & Gosha, 2018*) defined HS as "language that attacks or diminishes, that incites violence or hate against groups, based

on specific characteristics, such as physical appearance, religion, descent, national or ethnic origin, sexual orientation, gender identity, or others". HS is capable of occurring in different linguistic forms, even in subtly humorous ways. Due to the rapid evolution of social networking language, the majority of these attempts are still having trouble finding a workable answer (*Langham & Gosha, 2018*). Consequently, a thorough comprehension of the available literature is required.

Currently, there is a limited number of reviews and survey studies available on the topic of hate speech detection (*Al-Hassan & Al-Dossari, 2019*; *Alkomah & Ma, 2022*; *Rini, Utami & Hartanto, 2020*; *Schmidt & Wiegand, 2017*). For instance, *Schmidt & Wiegand (2017)* conducted a survey regarding the detection of hate speech. The author examines various domains that have been investigated for the automated identification of such offensive expressions using natural language processing. The survey offers a concise, thorough, and organized summary of the existing approaches and provides insights into automatic hate speech detection. However, the focus of the study is primarily on feature extraction, neglecting the examination of alternative detection methods and performance metrics. Additionally, the study fails to address the research challenges prevalent in the field or provide insights into potential future research directions. In a separate investigation, *Al-Hassan & Al-Dossari (2019)* examined the concept of hate speech, specifically focusing on "cyber hate" manifested through social media and the internet. The study further distinguished between various forms of antisocial behavior, such as cyberbullying, abusive and offensive language, radicalization, and hate speech. The author then presented a comprehensive examination of how text-mining techniques can be utilized in social networks. Additionally, the study explored specific challenges that could serve as guidance for developing an Arabic hate speech detection model. However, while the study encompassed Arabic data sources, it did not solely focus on the Twitter-sphere but rather considered a broader range of social media platforms and the internet sphere. Moreover, the study overlooked the categorization of Arabic language usage in composing hate speech. Furthermore, the study neglected to review the validation approaches employed in the hate speech identification method.

The rationale of our study is to conduct an extensive survey of various hate speech detection and classification algorithms. We recognize from the aforementioned review that existing surveys on hate speech detection might not have gathered the comprehensive information we got from various reputable academic data sources. Some earlier studies have been limited to a few academic sources, Arabic tweets datasets, and Arabic language categories, validation methods, overlooking the thorough examination of the advantages and disadvantages of different hate speech detection and classification systems. In addition, the majority of the mentioned studies adopted a formal literature review method, lacking research questions, search strategies, data extraction processes, and data analysis. As a result, there is a necessity for a more systematic approach to reviewing the current knowledge in HS identification studies in Arabic Twitter data.

The novelty of our work lies in using data from diverse and well-known academic sources to achieve our objective of identifying hate speech content on the Arabic Twitter dataset. Additionally, we have identified significant techniques, along with their respective

benefits and drawbacks, when applied to various Arabic dialect datasets concerning hate speech. Furthermore, we have covered deep learning and other crucial Artificial Intelligence (AI)-based hate speech identification approaches that have previously only been explored in limited investigations.

Concerning the audience this study is intended for, the comprehensive SLR aims to support academicians who are interested in identifying social media hate speech using AI techniques and addressing related issues. By utilizing this proposed SLR, researchers will have the ability to choose the most suitable identification and control mechanisms to combat hate speech effectively. Our work facilitates the comparison of numerous existing hate speech detection approaches in terms of their hate speech categories, feature engineering methods, performance metrics, and validation techniques employed. Additionally, this study will aid researchers in exploring current research opportunities, addressing concerns, and tackling challenges related to hate speech text feature extraction and classification techniques used by other researchers for hate speech text classification.

These are the significant contributions provided by this study:

1. A thorough investigation of the detection of hate speech on Arabic tweets dataset.
2. An outline of classification techniques employed in hate speech detection in Arabic tweets.
3. A comprehensive investigation of major performance metrics employed to evaluate and validate the hate speech identification performance in Arabic tweets.
4. A comprehensive investigation of various feature engineering techniques and validation methods to design algorithms for hate speech identification on Arabic tweets effectively.
5. List of some major challenges to research in this research area and identification of prospective future trends;

The rest of this systematic review is provided as follows: The "Research Methods" section explains the SLR's methodologies and procedures, including the research questions, search strategy, and selection criteria. Data extraction and synthesis of the selected studies are provided in the section "Data Extraction and Synthesis". The "Review Finding and Discussion" section discusses the findings from the study, while research challenges and the future direction of the research domain are provided in the "Research Challenges and Future Directions" section. Finally, the "Conclusion" section concludes the study.

## RESEARCH METHOD

PRISMA (Preferred Reporting Items for Systematic Reviews and Meta-Analyses), has been utilized in this study. It gives writers an organized framework to clearly and fully communicate the procedures and conclusions of their systematic evaluations. This systematic literature review study adheres to the *Kitchenham & Charters (2007)* standard. Planning, conducting, and reporting are the three main phases used in the SLR process. Decisions are made regarding the research topics and the protocol for the systematic search during the planning phase. The search protocol includes judgments about the

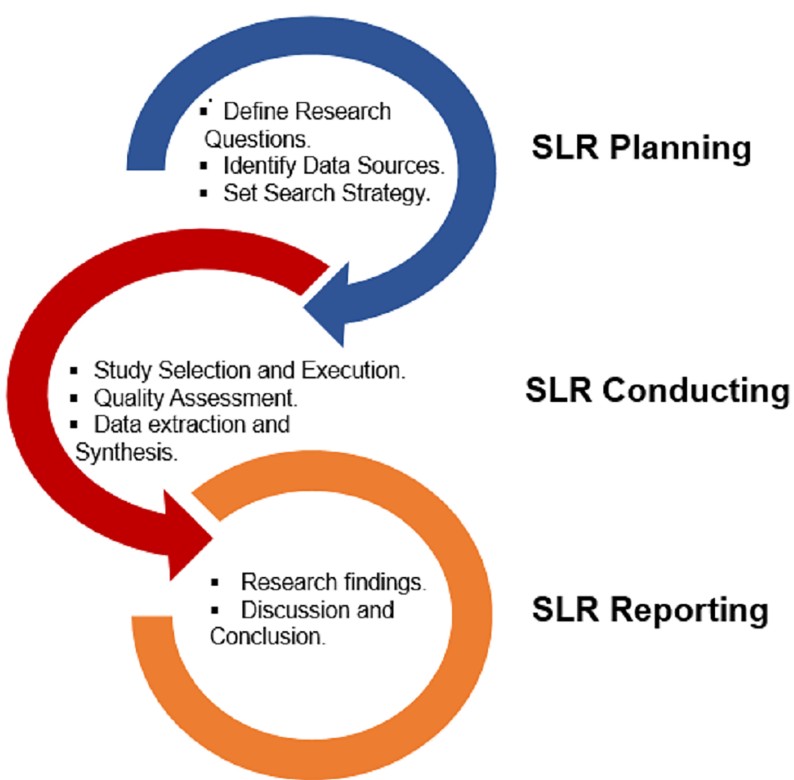

**Figure 1** **Research methodology summary.**

databases to be used, the search phrases to employ, and the selection criteria to use when looking up literature. In the second stage of the search method, the key selected articles undergo screening after collecting papers from the selected databases by employing the designated search criteria. With the help of search strings on the databases' keyword fields, the title and abstract are looked up. Only the studies that meet the criteria are gathered after the evaluation based on specified selection criteria. The suitable data needed to address the research questions is then collected from the selected papers. The summary of published articles is produced by combining the information. However, in the third and final stage, the research questions are addressed, and the conclusions are presented using summary tables and supplementary figures. The pictorial layout of the research methodology is depicted in Fig. 1.

## Planning the review

The first phase in an SLR process is review planning since it clarifies its objectives, identifies the need, determines how the review will be conducted, and explains why it is essential. During the planning phase of this SLR investigation, the following steps were taken:

a. Research question: It identifies research questions that will guide in performing the literature review.

b. Search strategy: It outlines the methods for compiling primary studies, by considering bibliographical databases and keywords.

c. Selection criteria: It provides information on the metrics chosen for the selection of the articles.

d. Quality assessment: It describes how the selected papers' quality is evaluated.

e. Data extraction and synthesis: It compares the selected publications and specified study topics.

## Research questions

The study aims to investigate recent developments in the field of HS identification to locate, assess, evaluate, and synthesize the research carried out on the topic of identifying hate speech on Twitter and to offer a summary of all the achievements made in the study of this subject. To conduct this SLR, the methods from *Kitchenham & Charters (2007)* were used. The current study was designed to provide answers to the following research questions (RQs).

RQ1: What type of Arabic language and hate speech categories are used in the selected studies?

RQ2: What identification techniques are used for Arabic HS identification on the Twitter dataset?

RQ3: What feature engineering techniques are commonly employed in HS identification for the Arabic Twitter dataset?

RQ4: What performance metrics are commonly used in HS identification for the Arabic Twitter dataset?

RQ5: What are the validation methods commonly employed in HS identification for the Arabic Twitter dataset?

## Search strategy

The evidence in finding the solution to hate speech identification on Arabic tweets was reviewed and summarized in this systematic literature review study. To get comprehensive results, both automatic and manual searches were performed. According to *Kitchenham (2004)*, one important aspect that distinguishes a systematic review from a conventional review is a rigorous search method (*Kitchenham & Charters, 2007*). The database search was completed in May 2023. In this systematic literature review investigation, the PRISMA standards were adhered to *Shamseer et al. (2015)*. The PICO (Participants, Intervention, Comparison, and Outcome) framework was used as a guide for the review, which includes:

Participants (P): Individuals that compose and post hate speech on social networking platforms, especially Twitter.

Interventions (I): The application of classification techniques in hate speech identification in Arabic tweets.

Comparators (C): To compare the hate speech and identification techniques that the scholarly literature has reported.

Outcome (O): Classification techniques and hate speech identification strategies that have been noted in the academic literature.

To carry out the literature search process, nine digital databases were used, including Science Direct, IEEE Explore, Taylor and Francis, ProQuest, MDPI, Springer, Hindawi, Sage, and Google Scholar. These online resources include the most significant peer-reviewed articles in the disciplines of computational and NLP techniques for Arabic text. By excluding data from citations and patent discoveries, this study analyzed the English-language scientific literature that was released between 2018 and 2023. The data search makes use of keywords, synonyms, related phrases, variants, or terms having the same meaning as the technologies. The next subsection gives more information about the search string.

## Search strings

To begin our SLR, we first identified the important keywords from the relevant research articles, as well as the research questions. Alternative synonyms, acronyms, and word spelling variants were added for the various keywords to enable a more thorough search. To identify as many publications as possible that were pertinent to the topic of interest, the search query was designed. Establishing the keywords was the first step, and the scientific community has so far used the term "hate speech" most of the time. This expression is the most often used to describe this category of damaging information created by users, which is even accepted as a legal term in certain jurisdictions (*Schmidt & Wiegand, 2017*). As a result, it is regarded as the phrase in this research subject that is more targeted in certain studies that have been published, the two additional synonyms "cyberhate" and "hateful language" have been employed as well. 'Cyber hatred' was used in *Cao, Lee & Hoang (2020)*, *Mozafari, Farahbakhsh & Crespi (2020)*, *Oak (2019)* and *Zhang, Robinson & Tepper (2018)*, whereas 'hateful language' was used in *Modi (2018)* and *Roesslein (2009)*. Following, we separated the keywords into three groups. Combining the keywords from each category resulted in the employment of the Boolean operator OR. The keywords were then included across all of the categories using the Boolean operator AND.

Query1 = ("Hate speech" OR "hateful language" OR "cyberhate" OR "cyberbullying" OR "offensive language" OR "Misogynistic" OR "religious hate")

Query2 = ("detection" OR "identification" OR "recognition" OR "classification")

Query3 = ("Arabic" OR "Arabic language" OR "Multilingual")

Thus,

Search Query = Query1 AND Query2 AND Query3

## Study selection criteria

To decide if a piece of writing belongs in the SLR or not, specific criteria were established. The most appropriate studies were found in the chosen studies using particular criteria for including and excluding articles. The papers gathered were thought to be the most closely comparable ones without duplicates. The criteria are chosen in advance to avoid the chance of bias (*Kitchenham & Charters, 2007*). The criteria for including and excluding

**Table 1 Inclusion criteria.**

| Inclusion no. | Criteria |
|---|---|
| Incl_1 | Papers about detecting Arabic hate speech that was published between 2018 and 2023. |
| Incl_2 | Only the primary study papers should be included |
| Incl_3 | Papers must be a peer-reviewed journals or conferences |
| Incl_4 | Papers must have been composed in the English language |
| Incl_5 | Papers involved hate speech detection from Twitter only and uses the Arabic dataset |
| Incl_6 | Papers that can directly answer one or more research questions |

**Table 2 Exclusion criteria.**

| Exclusion no. | Criteria |
|---|---|
| Excl_1 | The article did not carry out a hate speech detection study |
| Excl_2 | The article is not composed in the English language. |
| Excl_3 | The study collected data using various other social media channels like Facebook, YouTube, and Instagram. |
| Excl_4 | The article is not accessible. |
| Excl_5 | The review, proposal, or survey research study. |
| Excl_6 | The article was published earlier than 2018. |
| Excl_7 | The article used language different from Arabic for dataset collection. |

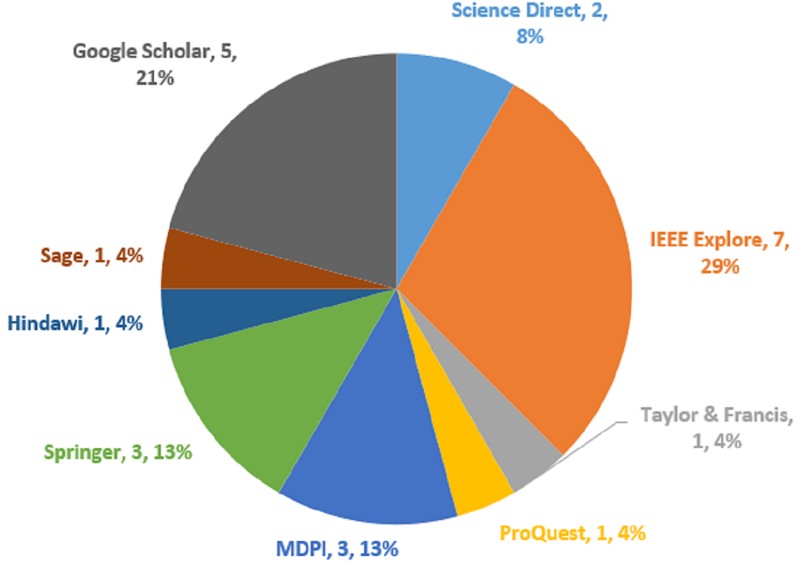

**Figure 2 Distributions of academic database and selected papers.**

papers were carefully designed to weed out any extracted papers that failed to fulfill the goals of the study. As a result, for an article to be included, every exclusion condition needs to be set to false whereas every exclusion condition needs to be set to true. Tables 1 and 2 depict the inclusion and exclusion criteria, whereas Fig. 2 shows the academic database distribution with the selected papers.

### Selection execution

A search is done to identify a list of studies that may be used for further analysis. The upkeep of the studies' bibliographies is handled *via* an endnote bibliography tool. After compiling a collection of related articles from the databases, search results were submitted to Endnote (*Peters, 2017*). Researchers were helped in their extensive evaluations of the literature about software engineering by the online resource endnote. The steps in the selection process are as follows:

1. Elimination of duplicates: Rarely are reports of comparable research reproduced, however, certain papers can be located in databases from multiple sources. During this initial stage, the endnote library automatically removes the replicated version.
2. Impurity eradication: In this subsequent filtering stage, the impurities from the search results are eliminated. The names of conferences related to the search phrases, for example, were included in the search results based only on the attributes of various electronic databases.
3. Title and abstract filtering: We checked the papers' titles and abstracts to discover the studies that mentioned and did not mention the search terms "hate speech identification," "Arabic language," and "Twitter."
4. Full-text filtering: At this stage, the full-text papers are downloaded and evaluated based on the study selection criteria.
5. Snowballing technique: To ensure that we capture as many relevant sources as we can, we used the forward and backward snowballing strategy (*Wohlin, 2014*). The process of locating papers using reference lists and citations is referred to as "snowballing," and it is an effective and trustworthy method for doing systematic literature reviews.

This systematic review coverage ranged from January 2018 to May 2023 and included all research articles that had been published during that time. Initially, 1,382 papers were retrieved, which consists of 235 papers from Science Direct, 340 papers obtained from IEEE Explore, 84 articles obtained from Taylor and Francis, 97 papers obtained from ProQuest, 120 papers obtained from MDPI, 162 papers obtained from Springer, 30 papers obtained from Hindawi, 25 papers obtained from Sage, and 289 papers obtained from Google Scholars. Out of the 1,382 articles that were filtered, duplicate content was detected in 76 of them. A total of 1,200 out of 1,382 articles were eliminated after reading the title and abstract. In addition, 82 articles were skipped during the last round of full-text reading and data extraction. A total of 24 articles in total were found to fulfill the inclusion criteria after the search. Table 3 depicts the research article identified, screened, and selected for this SLR study, whereas Fig. 3 represents the PRISMA flow diagram.

**Table 3 Research papers identified for SLR.**

| Digital databases | Initial query | Screening based on title and abstract | Screening based on a full reading |
|---|---|---|---|
| Science Direct | 235 | 22 | 2 |
| IEEE Explore | 340 | 27 | 7 |
| Taylor and Francis | 84 | 11 | 1 |
| ProQuest | 97 | 5 | 1 |
| MDPI | 120 | 3 | 3 |
| Springer | 162 | 3 | 3 |
| Hindawi | 30 | 1 | 1 |
| Sage | 25 | 1 | 1 |
| Google Scholar | 259 | 9 | 5 |
| Snowballing | 30 | 0 | 0 |
| Total | 1,382 | 106 | 24 |

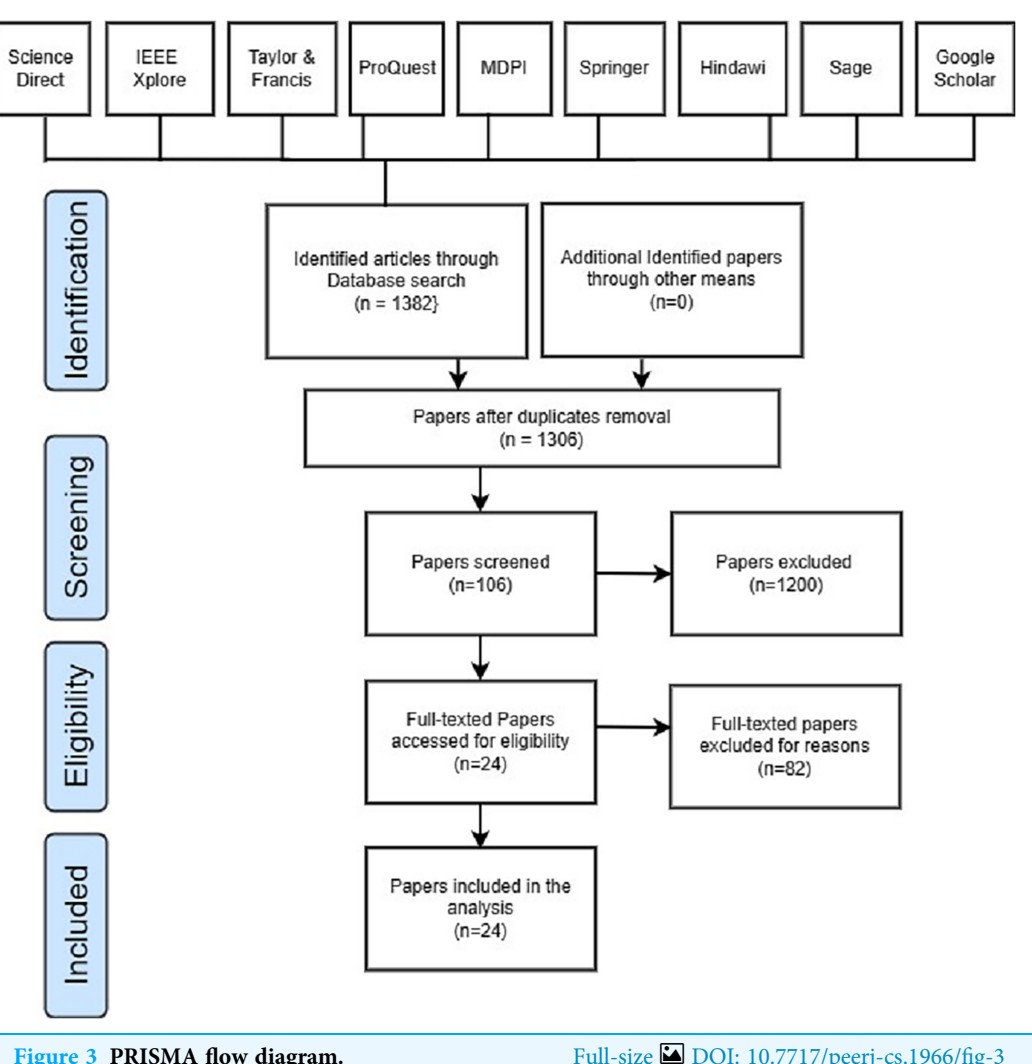

**Figure 3 PRISMA flow diagram.**

**Table 4 Data synthesis description.**

| S/N | Classes | Explanation |
|---|---|---|
| 1 | Paper ID (P-ID) | A unique number is assigned to a article |
| 2. | Author(s) | Name(s) of the author of the article |
| 3. | Year | Publication year of the article |
| 4 | Aim of the study | The goal for which the study was carried out |
| 4. | Classification technique | What classification techniques employed in Arabic Hate speech identification in Twitter data |
| 5. | Performance metrics | The performance metrics employed to evaluate the classification models for hate speech identification. |
| 7 | Databases | The database for sourcing the article. |
| 8. | Hate speech categories | The categories of hate speech that were researched. |
| 9. | Paper type | The types of the article (Journal article or conference proceedings) |
| 10. | Feature used | The features employed on the detection model for Arabic hate speech |
| 11. | Type of Arabic Language | The type of Arabic Language used in the hate speech dataset. |
| 12. | Validation type used | The type of validation used during the classification process. |

## Quality assessment

Every well-designed and carried-out SLR ought to contain a risk of bias evaluation, also known as a quality assessment (QA) of the studies (*Shamseer et al., 2015*). By scoring each selected study's merit following a set of standards, it is possible to determine the primary research's findings and their interpretation's importance (*Kitchenham & Charters, 2007*). A quality evaluation was carried out for this review to assess how relevant a study is and its potential to produce findings that will broaden the area of the investigation. The quality of each manuscript submitted for inclusion was evaluated at this stage. Evaluation of the papers' content value and applicability was the main objective. In this SLR article, the methodology used to assess the quality of each of the 24 papers included in this study is based on the study of *Kitchenham & Charters (2007)*. These standards are aimed at reassuring readers about the coverage of the SLR research questions rather than criticizing academic works (*Ouzzani et al., 2016*). Accordingly, the following quality evaluation inquiries were suggested.

QA1. Are the objectives and goals of the study clearly stated?

QA2: Has the study article's description of its inclusion and exclusion criteria been accurate?

QA3. Are the classification techniques in the research explained?

QA4. Do the study's findings advance the science of hate speech recognition in Arabic tweets?

Each quality assessment question was answered on a three-point scale: "Yes" received one point, "Partially" received 0.5 points, and "No" received 0 points. The article earns one point if it responds to the quality assessment question. If the quality assessment question is only partially answered, it receives 0.5 points. An article that failed to respond to the quality assessment question obtains a score of 0. By comparing the research papers' quality

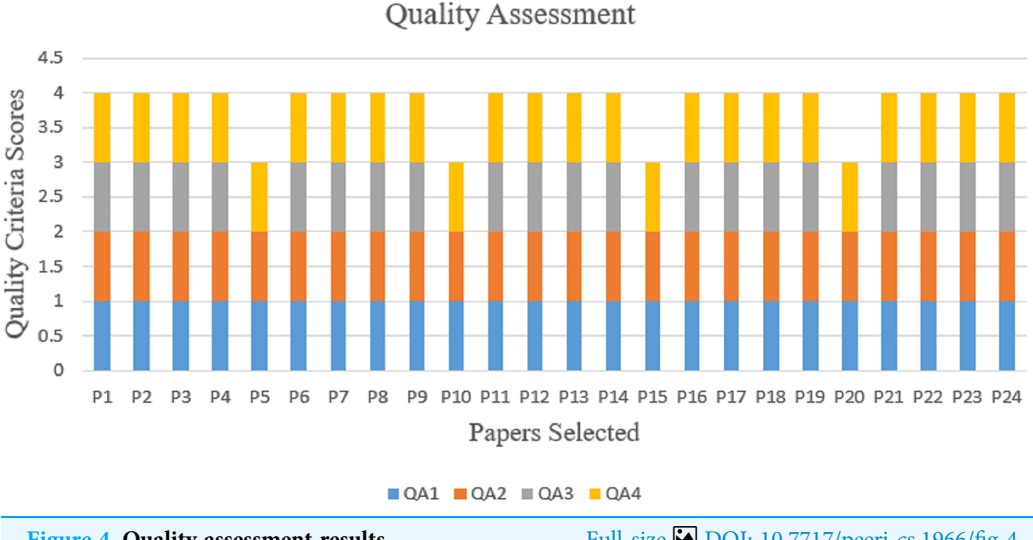

**Figure 4  Quality assessment results.**               

to the quality assessment questions, the research papers' quality was assessed. Table 4 lists the solutions to the QA questions for each article. Each research study had a total score that was determined. A cutoff point was established, such as if the final score was three or more, the research was added after that. On the other hand, the research study was excluded if it had fewer than three scores.

The quality assessment question (QA1) aims to evaluate whether the objectives and goal of the study are clearly articulated in the research article. The objectives and goal should provide a concise and specific description of what the study aims to achieve or investigate. Here, the review found 24 papers that match the first criteria.

The quality assessment question (QA2) pertains to the accuracy of the description of criteria for including and excluding the study article. Such criteria define the characteristics or factors used to determine participant or study eligibility. Accurate and precise reporting of inclusion and exclusion criteria ensures transparency and allows readers to evaluate the reliability and relevance of the study results. In this regard, the assessment identified 24 papers that match the second criterion.

The quality assessment question (QA3) focuses on the clarity and comprehensibility of the explanation provided for the classification techniques employed in the research. A well-explained classification technique allows readers to grasp the approach employed in the research and evaluate its appropriateness and effectiveness. In this assessment, 20 papers were identified that match the criteria.

The quality assessment question (QA4) assesses whether the study's findings contribute to the advancement of the science of hate speech (HS) recognition in Arabic Twitter data, which relies on the generation of new knowledge and the development of effective methods. In this assessment, 24 papers were also identified that match the criteria. Thus, it should be noted that the 24 papers that were assessed all met the criteria for the SLR purpose, and as a result, no papers were eliminated from this quality assessment phase Fig. 4 depicts the pictorial representation of the quality assessment evaluation results.

## DATA EXTRACTION AND SYNTHESIS

This systematic review research provides a thorough investigation of hate speech (HS) identification in Arabic Twitter datasets, research challenges, and open research directions. The process of data extraction is documented using a PRISMA flowchart. This process results in the creation of a data extraction form, which is then used to gather crucial information from the 24 articles (see Table 4). At this point, a data extraction form is made by carefully studying each of the 24 chosen articles to collect the information required to fulfill the study's aim and objectives, as indicated in Table 4. To record data from a few chosen articles, a standard information extraction form that was adapted from *Kitchenham & Brereton (2013)* recommendations was utilized. Using Endnote desktop software, the fundamental data has been structured, including "title, authors, publication date, Digital Object Identifier (DOI), and publication information", *etc*. The primary study was then used to obtain specific data from each article per the study classification. The data for the review was extracted from 12 columns in an MS Excel file: Paper ID (P-ID), Author(s), Year, the aim of the study, classification technique, performance metrics, Databases, hate speech categories, paper type, feature engineering technique used, type of Arabic language, and validation type used. The subsection 'selection execution' provided details on the period selected for the evaluation (2018–2023).

### Publication source

Table 5 depicts the 24 papers that were finally selected for the SLR studies, including 22 journal papers and two conference proceedings, from the field of research to perform this extensive systematic investigation.

### Publication year overview

The 24 publications selected between the 2018 and 2023 timeframes are shown in Fig. 5. The first primary study selected for this SLR study was published in the year 2018. According to the research for the study, there was a rise in publications between 2019 and 2021, as the trend in Fig. 5 demonstrates, which indicates the increasing interest in this study domain. However, the trend declined in the year 2022. Figure 5 further demonstrates that the vast majority of studies, with an average frequency of 8, were published in 2021. The analysis shows that at least one article was published each year for this literature study. However, the years 2018 and 2019 had an equal number of selected articles, which are three articles each. The analysis also showed that the least number of published articles was done in 2023, which could be that the year has not ended and there is a possibility of publishing more articles before the end of the year 2023. The majority of the examined articles were released during the last 3 years which indicates the increasing interest in this study domain.

### Publication coverage of research

The global research coverage areas are depicted in Fig. 6. As can be seen in Fig. 6, Saudi Arabia country provided the most articles (12), out of 24 selected articles for this study. Following that are Jordan with four studies, and Lebanon with two. Others such as

Table 5 Publication source distributions.

| S/N | Publication sources | Frequency |
| --- | --- | --- |
| 1 | Journal articles | 22 |
| 2 | Conference proceedings | 2 |

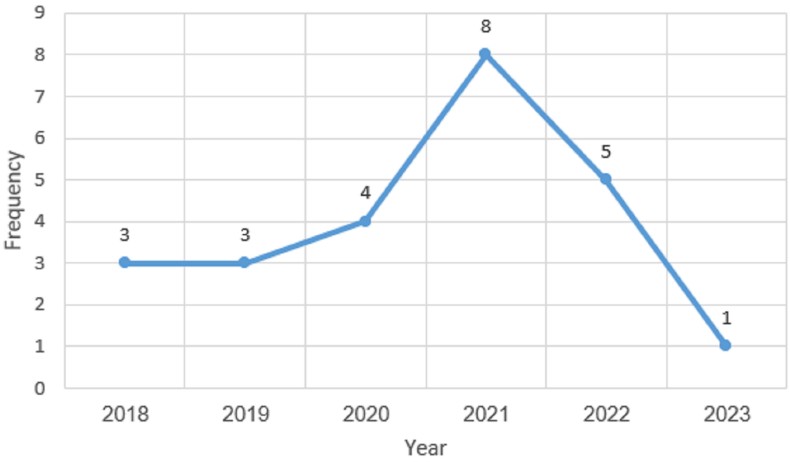

Figure 5 Publications year overview.

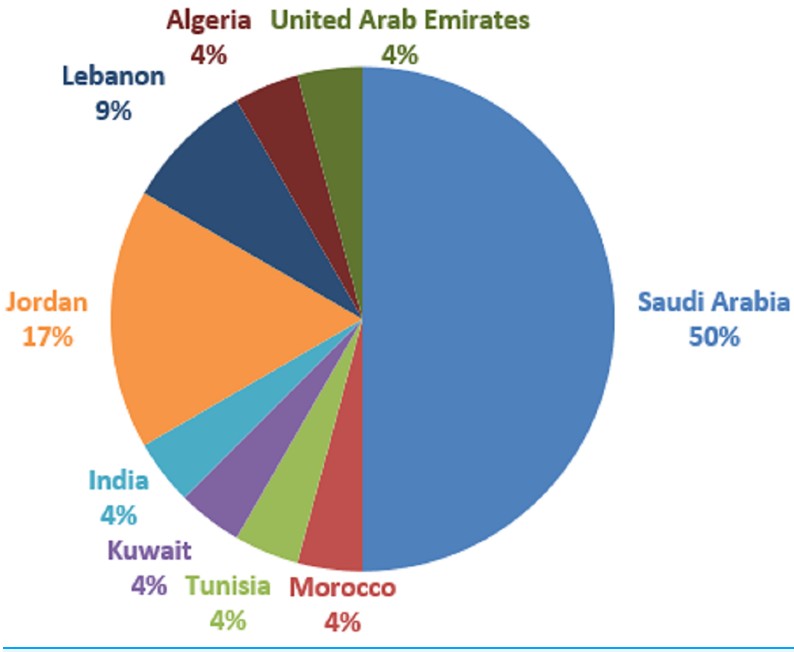

Figure 6 Publication coverage areas.

Morocco, Tunisia, Kuwait, India, Algeria, and the United Arab Emirates are having one study each. Based on our analysis, the coverage area shows that most Arabic hate speech is common in Western Asia. One of the reasons for such a coverage area is linguistic proficiency: These nations' official language is Arabic, and the researchers working there

are native Arabic speakers. They have an in-depth knowledge of Arabic, which is necessary for correctly identifying and deciphering hate speech in Arabic tweets. Local relevance is another reason. Although hate speech is a worldwide problem, it can have context-specific expressions. Because hate speech directly impacts these nations' communities and online spaces, researchers in these nations are compelled to confront the issue within their respective cultural, socioeconomic, and political settings. Besides, natural language processing (NLP), machine learning, and social media analysis are just a few of the many fields of research that are encouraged and supported by well-established academic and research institutions in these nations. Resources and financial possibilities are offered by these institutions to scholars. One other crucial reason is government and social concerns. Governments and civil society groups in these nations are concerned about hate speech on social media platforms. This worry may inspire research activities and projects targeted at detecting, reducing, or eliminating hate speech online. Data availability is another important reason. Researchers in these nations frequently have better access to Arabic tweet datasets since they may have local partners or sources for data collecting. To develop and evaluate hate speech detection models, relevant data access is essential.

## REVIEW FINDINGS AND DISCUSSIONS

The SLR's findings are presented in this section. After applying the research selection criteria, snowballing, and performing quality assessments, the databases produced 24 papers, which were chosen for further investigation. These 24 papers are listed in Table 6 along with details of their synthesis. Combining backward and forward snowballing led to the creation of a category for snowballing data sources. All chosen databases produced at least one article each. The next part addresses each of the five research questions listed in the "Research Method" section.

RQ1: What type of Arabic language and hate speech categories were used in the selected studies?

The results from the type of Arabic language and hate speech categories identified in the chosen papers show that the Arabic language is of different types and hate speech is of various categories.

Arabic is a Semitic language with different variations or forms across different regions and countries. There are two major categories into which Arabic can be generally divided; they are Classical Arabic and Modern Standard Arabic (MSA), also known as Literary Arabic. Additionally, there are numerous dialects of Arabic spoken in various countries and regions. In this study, there are four distinct varieties of Arabic spoken today, which are current standard Arabic, Levantine Arabic, Egyptian dialects Arabic, and Gulf dialectic Arabic, as explained below:

1. Modern Standard Arabic (MSA): MSA is a standardized type of Arabic based on Classical Arabic but modified to suit contemporary needs. It is used in written communication, formal speeches, media, and official documents across the Arab world. MSA serves as a lingua franca, enabling communication between Arabic speakers from different regions (*Duwairi, Hayajneh & Quwaider, 2021*).

Alhazmi et al. (2024), *PeerJ Comput. Sci.*, DOI 10.7717/peerj-cs.1966

**Table 6 Summary of the selected article synthesis.**

| P-ID | Aim of the study | Hate speech categories | Classification techniques (CT) | CT categories | Features used | Performance metrics | Type of Arabic language | Database | Paper type | Validation type used | Year | References |
|---|---|---|---|---|---|---|---|---|---|---|---|---|
| P1 | Deep learning approach Automatic Hate Speech Detection in the Saudi Twittersphere | Racist, Religious, Ideological, Tribal Hate speech | Deep learning (CNN, GRU & BERT) | ML-NLP | Continuous BOW | F1-SCORE, AUC | MSA | MDPI | Journal | Train/test | 2020 | Alshalan & Al-Khalifa (2020) |
| P2 | Offensive language detection using the BERT model | Offensive and non-offensive | BERT | NLP | TF-IDF | Accuracy, F1-score | MSA | Science Direct | Journal | Train/test | 2021 | El-Alami, El Alaoui & Nahnahi (2022) |
| P3 | Hate and Abusive language detection in Levantine | Hateful and non-hateful (Political) | SVM, NB | ML | N-gram. TF-IDF | Precision, Recall, F1-score, Accuracy | Levantine | Google Scholars | Conference | Train/test | 2019 | Mulki et al. (2019) |
| P4 | Combination of GRU NN with handcrafted features for religious hate speech detection in Arabic Twitter | Religious hate speech | Lexicon-based, SVM, LR, RNN | ML-NLP | N-gram, CBOW | Accuracy, Precision, Recall, F1-score, AUC | MSA | Springer | Journal | Train/test | 2018 | Albadi, Kurdi & Mishra (2019) |
| P5 | Impact of Preprocessing on Offensive Language Identification in Arabic | Offensive and non-offensive | SVM | NLP-ML | TF-IDF, character-based count vectorizer | Accuracy, precision, recall, and F1-score | MSA | Google Scholars | Conference | Train// Dev/test | 2020 | Husain (2020) |
| P6 | Misogyny and Sarcasm Detection in Arabic Texts | Misogynistic and non-Misogynistic | PAC LRC RFC LSVC DTC KNNC ARABERT | ML | BOW | Accuracy, precision, recall, F1-score | NS | Hindawi | Journal | Train/test | 2022 | Muaad et al. (2022) |

| P-ID | Aim of the study | Hate speech categories | Classification techniques (CT) | CT categories | Features used | Performance metrics | Type of Arabic language | Database | Paper type | Validation type used | Year | References |
|------|------------------|------------------------|-------------------------------|---------------|---------------|---------------------|-------------------------|----------|------------|----------------------|------|------------|
| P7 | Hate speech detection in Arabic social network. | Racism, religious, sexism, and political hate speech | SVM, DT, RF, NB | ML-NLP | BOW, TF, TF-IDF | Accuracy, precision, recall, G-mean | Levantine | Sage | Journal | 10-fold cross-validation | 2021 | Aljarah et al. (2021) |
| P8 | Detection of hate speech embedded in Arabic tweets. | Racial, Religious, Misogyny hate speech | CNN, CNN-LSTM, BiLSTM-CNN | ML | CBOW, SG | Accuracy, precision, recall, F1-score | MSA | Springer | Journal | Train/test | 2021 | Duwairi, Hayajneh & Quwaider (2021) |
| P9 | Detection of hate and offensive speech in Arabic social media | Racial, Religious, Gender | NB, SVM, LR, CNN, GRU, LSTM | ML | Word embeddings (Random, Skip-gram, Cbow, and Fasttext) and contextual word embedding (multilingual Bert) | Precision, Recall, F-macro | MSA, Gulf Arabic | Science Direct | Journal | Train/test | 2020 | Alsafari, Sadaoui & Mouhoub (2020) |
| P10 | Hate Speech Detection in Arabic. | Racial, Religious, Gender, Cyberbullying, Offensive and non-offensive | RNN | ML-NLP | CBOW, SG | Accuracy, precision, recall, F1-score | MSA | MDPI | Journal | Train/test | 2022 | Anezi (2022) |
| P11 | Religious Hate Speech Detection in the Arabic Twittersphere | Religious Hate Speech | Lexicon-based, LR SVM, GRU-based RNN. | ML | N-gram | Accuracy, Precision, Recall, F1-score, AUC | MSA | IEEE | Conference | Train/test | 2018 | Albadi, Kurdi & Mishra (2018) |
| P12 | Arabic Hate speech and Offensive Language detection in Twitter using BERT | Hateful, Offensive and non-hateful, non-Offensive | BERT, SVM, LR, sentiment analysis | ML-NLP | The text content of tweets, emojis, and sentiment analysis classification | Accuracy, Precision, Recall, F1-score, Micro-average | MSA | Proquest | Journal | Train/dev/test | 2022 | Althobaiti (2022) |

(Continued)

Alhazmi et al. (2024), *PeerJ Comput. Sci.*, DOI 10.7717/peerj-cs.1966

| P-ID | Aim of the study | Hate speech categories | Classification techniques (CT) | CT categories | Features used | Performance metrics | Type of Arabic language | Database | Paper type | Validation type used | Year | References |
|------|------|------|------|------|------|------|------|------|------|------|------|------|
| P13 | Cyberbullying Detection through Sentiment Analysis of Arabic Tweets | Cyberbullying | Lexicon-Based, SVM | ML-NLP | TF-IDF | Accuracy, Precision, Recall, F1-score | MSA | Google Scholar | Journal | Train/test | 2021 | *Almutairi & Al-Hagery (2021)* |
| P14 | Cyberbullying detection in Arabic | Cyberbullying | FFNN | ML-NLP | Word embedding | Accuracy | Levantine Arabic, Egyptian, Gulf | IEEE | Journal | Train/test | 2018 | *Haidar, Chamoun & Serhrouchni (2018)* |
| P15 | Deep learning detection of hate speech in Arabic tweets | Hateful and non-hateful | LTSM, CNN+LTSM, GRU, CNN+GRU | ML | Word embedding | Precision, Recall, F1-score | NS | Springer | Journal | 10 fold cross-validation | 2020 | *Al-Hassan & Al-Dossari (2022)* |
| P16 | Detection of Arabic Offensive Language in Microblogs using Word Embedding and Deep Learning | Offensive and non-offensive | BiLSTM | ML | CBOW | Precision, Recall, F1-score | MSA | IEEE | Journal | Train/test | 2022 | *Aljuhani, Alyoubi & Alotaibi (2022)* |
| P17 | Arabic Hate Speech Detection using semi-supervised learning | Hateful and non-hateful | CNN, BiLSTM | ML | N-gram | Precision, Recall, F1-score | MSA, Gulf Arabic dialect (GAD) | IEEE | Conference | Train/test | 2021 | *Alsafari & Sadaoui (2021a)* |
| P18 | Hate and Offensive Arabic Speech detection using semi-supervised learning. | Hateful and non-hateful | SVN CNN BILSTM | ML | Word-embedding | Precision, Recall, F1-score | MSA, Gulf Arabic dialect (GAD) | Taylor and Francis | Journal | Train/test | 2021 | *Alsafari & Sadaoui (2021b)* |

Alhazmi et al. (2024), *PeerJ Comput. Sci.*, DOI 10.7717/peerj-cs.1966

| P-ID | Aim of the study | Hate speech categories | Classification techniques (CT) | CT categories | Features used | Performance metrics | Type of Arabic language | Database | Paper type | Validation type used | Year | References |
|---|---|---|---|---|---|---|---|---|---|---|---|---|
| P19 | Offensive Language Detection in Arabic Social Networks | Offensive and non-offensive cyberbullying | KNN, NB, LR, DT, SVM, RF, XGBOOST | ML | Word-embedding, N-gram | Precision, Recall, F1-score, Accuracy | NS | IEEE | Journal | Train/test | 2022 | *Shannaq et al. (2022)* |
| P20 | Cyberbullying detection in Arabic on Social Networks Using ML | Cyberbullying | NB | ML | NIL | Precision, Recall, F-measure, Accuracy | NS | IEEE | Journal | Train/test | 2019 | *Mouheb et al. (2019)* |
| P21 | Offensive and Hate Speech Detection in Arabic Using a Cross-Corpora Multi-Task Learning Model | Hateful/ Offensive and non-hateful/ offensive | MTL-ARABERT, MTL-MARBERT | ML | CBOW, N-gram | Accuracy, F1-score | MSA | Google Scholar | Journal | Train/dev/test | 2021 | *Aldjanabi et al. (2021)* |
| P22 | Hate Speech Detection in Arabic using Word Embedding and Deep Learning. | Hateful and non-hateful | CNN, LSTM | ML-NLP | CBOW, SG | Precision, Recall, F-measure, Accuracy | NA | | Conference | Train/test | 2021 | *Faris et al. (2020)* |
| P23 | Cyberbullying in Arabic using ensemble learning | Cyberbullying | SVM, KNN, NB | ML | p-CBOW | Precision, Recall, F-measure | Levantine Arabic, Egyptian, Gulf | IEEE | Conference | NIL | 2019 | *Haidar, Chamoun & Serrhouchni (2019)* |
| P24 | Hate-Speech Detection in Arabic Social Media using BERT | Hateful and non-hateful | BERT | ML-NLP | Word-embedding | Precision, Recall, F1-score | NS | MDPI | Journal | cross-validation | 2023 | *Almaliki et al. (2023)* |

**Table 7 Types of Arabic language Identified in the selected papers.**

| Type of Arabic language | Frequency | References |
|---|---|---|
| MSA | 14 | *Anezi (2022)*, *Albadi, Kurdi & Mishra (2018)*, *Althobaiti (2022)*, *Almutairi & Al-Hagery (2021)*, *Aljuhani, Alyoubi & Alotaibi (2022)*, *Alsafari & Sadaoui (2021a, 2021b)*, *Aldjanabi et al. (2021)*, *Alshalan & Al-Khalifa (2020)*, *El-Alami, El Alaoui & Nahnahi (2022)*, *Albadi, Kurdi & Mishra (2019)*, *Husain (2020)*, *Duwairi, Hayajneh & Quwaider (2021)*, *Alsafari, Sadaoui & Mouhoub (2020)* |
| Levantine | 4 | *Haidar, Chamoun & Serhrouchni (2018, 2019)*, *Mulki et al. (2019)*, *Aljarah et al. (2021)* |
| Gulf | 5 | *Alsafari & Sadaoui (2021a, 2021b)*, *Alsafari, Sadaoui & Mouhoub (2020)*, *Haidar, Chamoun & Serhrouchni (2019, 2018)* |
| Egyptian | 2 | *Haidar, Chamoun & Serhrouchni (2018, 2019)* |
| No mention | 6 | *Al-Hassan & Al-Dossari (2022)*, *Shannaq et al. (2022)*, *Mouheb et al. (2019)*, *Faris et al. (2020)*, *Almaliki et al. (2023)*, *Muaad et al. (2022)* |

2. Egyptian Arabic: This type of Arabic is being spoken in Egypt and is widely understood across the Arab world due to Egypt's prominent media industry (*Haidar, Chamoun & Serhrouchni, 2019*).

3. Levantine Arabic: Lebanon, Syria, Jordan, and Palestine speak this type of Arabic dialect. This dialect is further divided into sub-dialects such as Lebanese, Syrian, and Palestinian (*Aljarah et al., 2021*).

4. Gulf Arabic: This type is being spoken throughout the Gulf Cooperation Council (GCC), which includes Saudi Arabia, the United Arab Emirates, Qatar, Bahrain, Kuwait, and Oman. Its distinctive vocabulary and pronunciation serve as defining characteristics (*Alsafari & Sadaoui, 2021a*).

It should be noted that Egyptian Arabic, Levantine Arabic, and Gulf Arabic are classified as Arabic Dialects. Arabic dialects refer to the regional varieties of Arabic spoken in different countries and regions. These dialects have significant linguistic variations in terms of pronunciation, vocabulary, and grammar. Each dialect has its unique characteristics, influenced by the historical, cultural, and linguistic factors specific to its region. It is important to note that while Modern Standard Arabic (MSA) is the formal standard, it is not a natively spoken language and is primarily used in formal settings, while dialects are used in daily conversations and informal contexts within their respective regions.

Table 7 illustrates the type of Arabic language identified in the literature. As can be seen, MSA was identified in 14 out of 24 selected studies, followed by Gulf dialects Arabic found in five studies, followed by Levantine identified in five studies, and Egyptian dialects identified in two studies. However, five studies out of 24 studies did not mention the type of Arabic language that was used in the dataset. The pictorial representation is indicated in Fig. 7.

On the other hand, HS refers to any kind of conversation, whether spoken, written, or symbolic, that discriminates, threatens, or incites violence or activities that are unfair to individuals or groups based on protected traits including race, ethnicity, religion, gender,
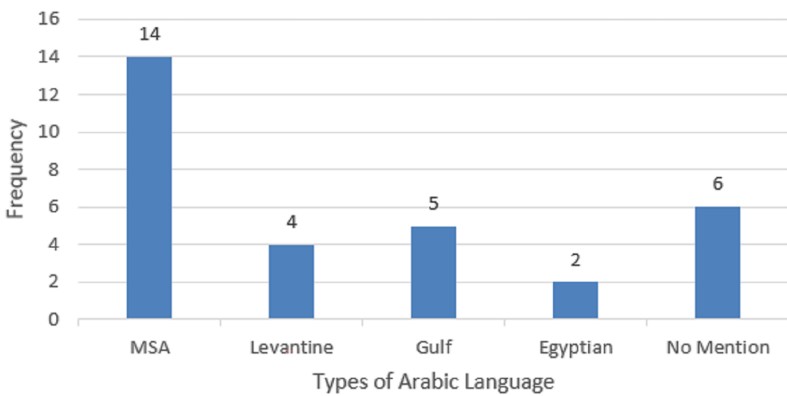

**Figure 7** Pictorial representation of Arabic language categories.

**Table 8** Hate speech categories.

| Hate speech categories | Frequency | References |
|---|---|---|
| Religious hate speech | 6 | *Alshalan & Al-Khalifa (2020)*, *Albadi, Kurdi & Mishra (2019)*, *Aljarah et al. (2021)*, *Alsafari & Sadaoui (2021b)*, *Anezi (2022)*, *Albadi, Kurdi & Mishra (2018)*. |
| Racism | 5 | *Alshalan & Al-Khalifa (2020)*, *Anezi (2022)*, *Alsafari & Sadaoui (2021b)*, *Aljarah et al. (2021)*, *Duwairi, Hayajneh & Quwaider (2021)*. |
| Offensive | 5 | *El-Alami, El Alaoui & Nahnahi (2022)*, *Husain (2020)*, *Aljuhani, Alyoubi & Alotaibi (2022)*, *Shannaq et al. (2022)*, *Aldjanabi et al. (2021)*. |
| Cyberbullying hate speech | 4 | *Almutairi & Al-Hagery (2021)*, *Haidar, Chamoun & Serhrouchni (2018)*, *Shannaq et al. (2022)*, *Mouheb et al. (2019)*. |
| Sexism and Misogyny | 3 | *Muaad et al. (2022)*, *Aljarah et al. (2021)*, *Duwairi, Hayajneh & Quwaider (2021)*. |
| Ideological hate speech | 1 | *Alshalan & Al-Khalifa (2020)*. |
| Tribal hate speech | 1 | *Alshalan & Al-Khalifa (2020)*. |
| Political hate speech | 1 | *Mulki et al. (2019)*. |

sexual orientation, nationality, or handicap. While HS can take various forms and is highly context-dependent, the following 10 categories of Arabic hate speech were identified in this study, including hateful, offensive, racial hate, religious hate, ideology hate, tribal hate, political hate, sexism, and misogyny, gender hate, and cyberbullying. Table 8 illustrates the summary of the hate speech categories. A brief definition of each category is provided below:

1. Racial or ethnic hatred: Racial hate speech refers to expressions or acts that target individuals or groups based on their race or ethnicity and aim to discriminate against, incite hatred, or provoke hostility towards them. It involves using language or actions that promote prejudice, racism, or discrimination based on racial grounds. Hate speech targeting a specific race or ethnic group, is often characterized by derogatory slurs,

stereotypes, or dehumanizing language (*Duwairi, Hayajneh & Quwaider, 2021*). Racial hate speech perpetuates racism, contributes to social divisions, and undermines inclusivity and equality. It is important to promote understanding, respect, and tolerance among diverse racial and ethnic groups and work toward eradicating racial discrimination.

2. Religious hate speech: Religious hate speech refers to expressions or acts that target individuals or groups based on their religious beliefs or affiliations and aim to discriminate against, incite hatred, or provoke hostility towards them. It involves using language or actions that promote prejudice, intolerance, or discrimination based on religious grounds (*Albadi, Kurdi & Mishra, 2018*). This can include vilification, mocking, or denigration of religious practices, symbols, or figures. Religious hate speech can contribute to religious intolerance, fuel conflicts, and undermine social harmony and coexistence. It is important to promote interfaith dialogue, respect for diversity, and understanding among individuals of different religious backgrounds.

3. Sexism and misogyny: Sexism and misogyny hate speech refers to expressions or acts that target individuals or groups based on their gender, particularly women, and aim to discriminate against, demean, or provoke hostility towards them (*Duwairi, Hayajneh & Quwaider, 2021*; *Waseem & Hovy, 2016*). It involves using language or actions that promote prejudice, inequality, or discrimination based on gender. This hate speech devalues, objectifies, or discriminates against individuals based on their gender. It may involve sexist slurs, derogatory comments, or the promotion of gender-based stereotypes. Sexism and misogyny hate speech perpetuate gender inequality, contribute to the marginalization of women, and undermine efforts to achieve gender equity and inclusivity. It is important to challenge and address sexist attitudes, promote gender equality, and create safe and respectful environments for all individuals.

4. Offensive hate speech refers to speech or expression that is not only discriminatory or prejudiced but also goes beyond that to deliberately provoke, insult, or demean individuals or groups based on their protected characteristics (*Aldjanabi et al., 2021*). It aims to offend, humiliate, or incite hostility toward the targeted individuals or communities. The offensive nature of hate speech lies in its intent to inflict harm, perpetuate stereotypes, and undermine the dignity and rights of the targeted individuals or groups (*Saeed, Calders & Kamiran, 2020*). Offensive hate speech can take many forms, including derogatory slurs, explicit insults, dehumanizing language, graphic imagery, or explicit calls for violence or discrimination. It often seeks to spread fear, division, and animosity within society, creating a hostile and toxic environment for those targeted

5. Ideological hate speech refers to expressions or acts that target individuals or groups based on their ideological beliefs, political affiliations, or social ideologies (*Alshalan & Al-Khalifa, 2020*). It involves using language or actions to discriminate against, incite hatred, or provoke hostility towards individuals or groups based on their ideological stance. Examples of ideological hate speech can include demonization and dehumanization, Stereotyping and generalizations, Incitement to violence or

discrimination, Verbal attacks and insults, Propagation of conspiracy theories, Intolerance, and exclusion It is required to note that freedom of speech and the boundaries of ideological criticism can vary across jurisdictions and cultural contexts. However, hate speech that targets individuals or communities based on their ideological beliefs often aims to create division, foster hostility, and undermine the principles of mutual respect and tolerance.

6. Tribal hate speech refers to expressions or actions that target individuals or groups based on their tribal or clan affiliations and aim to discriminate against, incite hatred, or provoke hostility toward them. It involves using derogatory language, stereotypes, insults, or promoting prejudice and hostility based on tribal grounds (*Alshalan & Al-Khalifa, 2020*). Examples of tribal hate speech can include Insults and derogatory language, Stereotyping and generalizations, Incitement to violence or discrimination, *etc*. Tribal hate speech can have severe consequences, including escalating inter-tribal tensions, inciting violence, and undermining social cohesion within communities. It is important to promote understanding, respect, and dialogue among different tribal groups to foster harmony and peaceful coexistence.

7. Political hate speech refers to expressions or acts that target individuals or groups based on their political beliefs, affiliations, or ideologies and aim to discriminate against, incite hatred, or provoke hostility toward them (*Guellil et al., 2020*). It involves using language or actions that promote prejudice, animosity, or discrimination based on political grounds. Examples of political hate speech can include Personal attacks and insults, Verbal harassment, intimidation, *etc*. Hate speech that targets individuals or communities based on their political beliefs often aims to create division, foster hostility, and undermine constructive dialogue and democratic principles.

8. Gender hate speech refers to expressions or acts that target people or groups according to their gender or gender identity and aim to discriminate against, incite hatred, or provoke hostility towards them. It involves using language or actions that promote prejudice, misogyny, transphobia, or discrimination based on gender (*Duwairi, Hayajneh & Quwaider, 2021*). Examples of gender hate speech can include, Sexist slurs and derogatory language, *etc*. Gender hate speech perpetuates inequality, fosters discrimination, and undermines efforts to achieve gender equality and inclusivity. It is important to promote respectful dialogue, challenge harmful stereotypes, and advocate for equal rights and opportunities for all genders.

9. Cyberbullying hate speech refers to using digital communication tools like social media, chat rooms, or messaging applications to target and harass people or groups with offensive, demeaning, or harmful language or content. It involves using technology to spread hate, incite hostility, or provoke emotional distress toward others (*Omar, Mahmoud & Abd-El-Hafeez, 2020*). Examples of cyberbullying hate speech can include; Harassment and insults, Threats and intimidation

RQ2: What identification techniques are used for Arabic HS identification on the Twitter dataset?

**Table 9 Detection techniques categories.**

| Detection techniques categories | References |
|---|---|
| Machine learning | *Mulki et al. (2019)*, *Muaad et al. (2022)*, *Aljarah et al. (2021)*, *Duwairi, Hayajneh & Quwaider (2021)*, *Alsafari, Sadaoui & Mouhoub (2020)*, *Albadi, Kurdi & Mishra (2018)*, *Al-Hassan & Al-Dossari (2022)*, *Aljuhani, Alyoubi & Alotaibi (2022)*, *Alsafari & Sadaoui (2021a, 2021b)*, *Shannaq et al. (2022)*, *Mouheb et al. (2019)*, *Aldjanabi et al. (2021)*, *Haidar, Chamoun & Serhrouchni (2019)*, |
| Natural language processing | *El-Alami, El Alaoui & Nahnahi (2022)* |
| Machine learning-natural language processing | *Alshalan & Al-Khalifa (2020)*, *Albadi, Kurdi & Mishra (2019)*, *Husain (2020)*, *Anezi (2022)*, *Althobaiti (2022)*, *Almutairi & Al-Hagery (2021)*, *Haidar, Chamoun & Serhrouchni (2018)*, *Faris et al. (2020)*, *Almaliki et al. (2023)* |

In the studies reviewed, the detection techniques to identify Arabic hate speech on Twitter in all of the research that was chosen. The detection techniques in the selected studies can be divided into three primary types: machine learning, NLP, and a combination of these two types of processing. The categorization methods used in the selected articles on hate speech identification in Arabic tweets are summarized in Table 9. The description of each of the categories is provided below:

(1) Machine learning techniques (ML) technique: ML is an aspect of artificial intelligence (AI) that makes it possible for systems to learn and enhance their performance through experience, without requiring explicit programming (*Janiesch, Zschech & Heinrich, 2021*). It involves the computer's ability to self-teach decision-making by utilizing training data. Machine learning is particularly valuable for tackling complex tasks that are impractical to code manually, such as addressing issues related to cyberbullying (*Haidar, Chamoun & Serhrouchni, 2018*). When faced with new data, the computer applies supervised or unsupervised learning algorithms for classification (*Chauhan et al., 2021*). In supervised learning, the algorithms rely on labeled training data, where inputs are provided along with corresponding labels. The algorithm learns by comparing its output with the labeled data, identifying errors, and adjusting the model accordingly. On the other hand, unsupervised learning algorithms operate on data without historical labels or using non-labeled data. In this case, the algorithm must discern patterns and structures within the data to understand its content. This process aids the system in classifying the data based on similarities and differences observed among the data points (*Chauhan et al., 2021*).

(2) Natural language processing (NLP) technique: Modified NLP empowers computers to comprehend and execute instructions given in natural language. These techniques facilitate the reading and understanding of text by leveraging linguistic knowledge. NLP techniques offer the ability to determine various features, such as semantic and syntax features, which play a crucial role in hate speech detection methods and differentiate between different approaches (*Albadi, Kurdi & Mishra, 2019*; *Schmidt & Wiegand, 2017*). In their work, *Devlin et al. (2018)* presents bidirectional encoder representations from transformers. BERT as it is known, is a brand-new model for representing languages. BERT has proven to perform exceptionally well on a variety of natural language processing jobs, such as sentiment analysis, question-answering, and textual entailment. However,

there has not been much research done on how well BERT works to identify hate speech in Arabic. Employing ML and data mining techniques, the natural language processing approach can effectively manage vast volumes of data to extract insightful information. However, it is crucial to carry out essential preprocessing procedures when working with textual documents to extract numerical and statistical information from the textual data, enabling their application in machine learning algorithms. Any text preprocessing and structuring solution must incorporate ideas like tokenization, stop-word elimination, and text vectorization.

(3) Machine learning and natural language processing (ML-NLP) techniques: The combination of ML and NLP offers several benefits, including automatic attribute identification (*Al-Makhadmeh & Tolba, 2020*). In the context of automatic classification, feature extraction plays a crucial role. NLP techniques extract features from the text, which are then analyzed by machine learning classifiers to identify patterns. Furthermore, the aim, utilized feature engineering techniques, hate speech categories, types of Arabic language, and performance measures of each study were outlined in Table 6. Hate speech detection employs various classification techniques depending on the specific problem being addressed. Some studies even employ hybrid, ensemble, or comparative approaches, which combine different algorithms or incorporate techniques from other domains into ML workflows. Ensemble models utilize multiple learning algorithms, leading to improved predictive outcomes compared to individual algorithms. The main difference between ensemble and hybrid approaches is that ensemble methods independently vote on the outcome, whereas hybrid methods predict a single conclusion without taking voting into account. The analysis of the chosen papers showed that classic classifiers such as logistic regression (LR), random forests (RF), support vector machines (SVM), and naive Bayes (NB) were most frequently utilized. These algorithms, proven useful in text classification through recent comparative studies, rely on appropriate feature selection for successful application (*Aljarah et al., 2021*). In addition to selecting optimal feature extraction methods, considering the data architecture (*Suhaidi, Kadir & Tiun, 2021*) and combining different feature selection approaches (*Oskouei & Razavi, 2018*) can enhance classification results. On the other hand, deep learning techniques eliminate the need for handcrafted features. Deep learning has gained significant popularity for HS identification in Arabic Twitter data since 2017 (*Badjatiya et al., 2017*), primarily due to its capacity to research classification appropriate to data representations (*Husain, 2020*; *Mansur, Omar & Tiun, 2023*). Well-known deep learning techniques include CNNs and LSTM networks (*Duwairi, Hayajneh & Quwaider, 2021*). CNNs are proficient in extracting contextual features and delivering modern outcomes for challenges including text, audio, video, and image categorization. LSTM networks, as a unique type of RNN, leverage central memory to capture dependence over time. Another type of RNN called GRU (*Alshalan & Al-Khalifa, 2020*), exploits its gating technique to learn words and how to connect long distances. Information is processed both forward and backward in BiLSTM. However, due to various structural arrangements, deep learning model performance differed among investigations. Several studies incorporated deep learning techniques, while others used

just one. multiple approaches such as CNN and GRU (*Al-Hassan & Al-Dossari, 2022*) or CNN and Bi-LSTM (*Arango, Pérez & Poblete, 2019*) within a single model. Recent advancements include the utilization of pre-trained BERT (bidirectional encoder representations from transformers) models, which produced better outcomes in the identification of hate speech (*El-Alami, El Alaoui & Nahnahi, 2022*; *Li et al., 2021*). Conversely, a few studies applied a hybrid approach, combining machine learning with lexicon-based techniques (*Albadi, Kurdi & Mishra, 2018*; *Almutairi & Al-Hagery, 2021*). A method based on lexicons, also referred to as lexicon-based sentiment analysis or dictionary-based sentiment analysis, is an NLP approach that determines sentiment or polarity in the text by utilizing pre-built lexicons or dictionaries containing words or phrases annotated with sentiment scores. For example, *Albadi, Kurdi & Mishra (2018)* conducted research on HS identification in Arabic tweets using a combination of lexicon and machine learning techniques. They created the first Arabic dataset of tweets with religious hate speech annotations that were made accessible to the public. In addition, the writers produced three publicly accessible Arabic lexicons including hate indexes and terms relating to religion. They thoroughly examined the labeled information, communicating the most often mentioned religious communities in both hateful and non-hateful tweets according to the nation of origin. Seven learning algorithms were trained on the labeled dataset using lexicon-based, n-gram-based, and deep learning techniques. These models were then tested on a fresh, previously unexplored dataset to determine how well the algorithms could generalize., achieving a promising result.

RQ3: What feature engineering techniques are commonly employed in HS identification for the Arabic Twitter dataset?

In hate speech detection, feature extraction plays a crucial role in extracting pertinent and distinctive information from the hate speech dataset. This extracted information aids in training the hate speech identification model. The review of the chosen papers revealed that the majority of them concentrated on leveraging the semantic properties of sentence features. Additionally, to extract linguistic and content-based attributes, researchers used automatic feature extraction approaches. Content-based features in text classification refer to the characteristics and information extracted from the textual content itself. These features capture the inherent properties and patterns within the text that can be utilized to distinguish and classify different categories or classes. Content-based features can be derived from various techniques and representations of the text. This was accomplished using various algorithms and statistical methods. The methods for extracting features from content used in the studies included Bag of the Word (BoW) (*Muaad et al., 2022*), Continuous Bag-of-Word (CBOW) (*Alshalan & Al-Khalifa, 2020*), term frequency and inverse document frequency (TF-IDF) (*El-Alami, El Alaoui & Nahnahi, 2022*), word embedding (*Alsafari, Sadaoui & Mouhoub, 2020*) and n-gram (*Mulki et al., 2019*). Table 10 illustrates that the majority of the selected studies employed the n-gram, C-BOW, and word embedding feature extraction technique with nine, eight, and six studies respectively. N-grams represent contiguous sequences of n-words in the text. Unigrams, bigrams, and trigrams, for instance, each represent a single word, a pair of words, and a triplet of words, respectively. N-grams capture local patterns and dependencies in the text. For

**Table 10 Feature engineering techniques used in the selected papers.**

| S/N | Features used | Paper ID |
|---|---|---|
| 1 | N-gram | *Mulki et al. (2019)*, *Albadi, Kurdi & Mishra (2019, 2018)*, *Althobaiti (2022)*, *Almutairi & Al-Hagery (2021)*, *Al-Hassan & Al-Dossari (2022)*, *Muaad et al. (2022)*, *Alsafari & Sadaoui (2021b)*, *Aldjanabi et al. (2021)* |
| 2 | C-BOW | *Alshalan & Al-Khalifa (2020)*, *Albadi, Kurdi & Mishra (2019)*, *Duwairi, Hayajneh & Quwaider (2021)*, *Anezi (2022)*, *Albadi, Kurdi & Mishra (2018)*, *Aljuhani, Alyoubi & Alotaibi (2022)*, *Faris et al. (2020)*, *Haidar, Chamoun & Serhrouchni (2019)* |
| 3 | Word-embedding | *Alsafari, Sadaoui & Mouhoub (2020)*, *Al-Hassan & Al-Dossari (2022)*, *Al-Hassan & Al-Dossari (2022)*, *Alsafari & Sadaoui (2021b)*, *Shannaq et al. (2022)*, *Almaliki et al. (2023)* |
| 4 | TF-IDF | *El-Alami, El Alaoui & Nahnahi (2022)*, *Mulki et al. (2019)*, *Husain (2020)*, *Aljarah et al. (2021)*, *Almutairi & Al-Hagery (2021)* |
| 5 | SG | *Duwairi, Hayajneh & Quwaider (2021)*, *Anezi (2022)*, *Faris et al. (2020)* |
| 6 | BOW | *Muaad et al. (2022)*, *Aljarah et al. (2021)* |

example, authors such as *Aldjanabi et al. (2021)*, *Al-Hassan & Al-Dossari (2022)*, and *Albadi, Kurdi & Mishra (2018)*, utilized the n-gram method for feature extraction in identifying hate speech in Arabic tweets. They claimed that the n-gram method works well for obtaining lexical features The simplicity and scalability properties of the n-gram model were among the reasons for its usage by these researchers, as it could handle large sample datasets.

CBOW is a well-liked technique for word embedding training in NLP tasks. Predicting a target word from its context words is the goal of CBOW. On either side of the target word, there is often a predetermined window size that defines the context words. The model uses a hidden layer to attempt to predict the target word using the input words from the context. The hidden layer acts as an encoding layer that captures the distributed representation of the context words. The CBOW technique was employed by the author in *Haidar, Chamoun & Serhrouchni (2019)* to extract the discriminative feature for Arabic HS identification. The study revealed the effectiveness of such a feature in improving the model performance. Word embeddings represent words as dense vector depictions, capturing semantic relationships between words. Techniques like Word2Vec, GloVe, or FastText are commonly used to generate word embeddings. For instance, a study by *Haidar, Chamoun & Serhrouchni (2018)* tested the performance of various word embeddings for Arabic hate speech identification using a deep learning model, and the performance indicated the effectiveness of the word embeddings in capturing semantic relatedness in the hate speech text.

The performance of classification can be affected by various factors, including the dataset's volume, selected features, and the uniqueness of classes (*Sajjad et al., 2019*). Relying on a single feature to handle a variety of data and discrepancies is not advised. Therefore, it is crucial to carefully choose an appropriate combination of features (*Sajjad et al., 2019*). For instance, authors (*Alsafari & Sadaoui, 2021b*) recognized the importance of feature fusion in cyberbullying identification. In their study, the author merged N-Grams, World2Vec, Skip-Gram, AraBert, and DistilBert for offensive and hate speech identification in social media platforms. Table 10 presents the feature engineering

**Table 11 Performance metric utilized in the selected studies.**

| S/N | Performance metrics used | Paper ID |
|---|---|---|
| 1 | F1-score | *Alshalan & Al-Khalifa (2020)*, *El-Alami, El Alaoui & Nahnahi (2022)*, *Mulki et al. (2019)*, *Albadi, Kurdi & Mishra (2019)*, *Husain (2020)*, *Muaad et al. (2022)*, *Duwairi, Hayajneh & Quwaider (2021)*, *Anezi (2022)*, *Albadi, Kurdi & Mishra (2018)*, *Althobaiti (2022)*, *Almutairi & Al-Hagery (2021)*, *Al-Hassan & Al-Dossari (2022)*, *Al-Hassan & Al-Dossari (2022)*, *Aljuhani, Alyoubi & Alotaibi (2022)*, *Alsafari & Sadaoui (2021a, 2021b)*, *Shannaq et al. (2022)*, *Mouheb et al. (2019)*, *Aldjanabi et al. (2021)*, *Faris et al. (2020)*, *Haidar, Chamoun & Serhrouchni (2019)*, *Almaliki et al. (2023)* |
| 2 | Precision | *Mulki et al. (2019)*, *Albadi, Kurdi & Mishra (2019)*, *Husain (2020)*, *Muaad et al. (2022)*, *Aljarah et al. (2021)*, *Duwairi, Hayajneh & Quwaider (2021)*, *Alsafari, Sadaoui & Mouhoub (2020)*, *Anezi (2022)*, *Albadi, Kurdi & Mishra (2018)*, *Althobaiti (2022)*, *Almutairi & Al-Hagery (2021)*, *Al-Hassan & Al-Dossari (2022)*, *Aljuhani, Alyoubi & Alotaibi (2022)*, *Alsafari & Sadaoui (2021a, 2021b)*, *Shannaq et al. (2022)*, *Mouheb et al. (2019)*, *Faris et al. (2020)*, *Haidar, Chamoun & Serhrouchni (2019)*, *Almaliki et al. (2023)* |
| 3 | Recall | *Mulki et al. (2019)*, *Albadi, Kurdi & Mishra (2019)*, *Husain (2020)*, *Muaad et al. (2022)*, *Aljarah et al. (2021)*, *Duwairi, Hayajneh & Quwaider (2021)*, *Alsafari, Sadaoui & Mouhoub (2020)*, *Anezi (2022)*, *Albadi, Kurdi & Mishra (2018)*, *Althobaiti (2022)*, *Almutairi & Al-Hagery (2021)*, *Al-Hassan & Al-Dossari (2022)*, *Aljuhani, Alyoubi & Alotaibi (2022)*, *Alsafari & Sadaoui (2021a, 2021b)*, *Shannaq et al. (2022)*, *Mouheb et al. (2019)*, *Faris et al. (2020)*, *Haidar, Chamoun & Serhrouchni (2019)*, *Almaliki et al. (2023)* |
| 4 | Accuracy | *Alshalan & Al-Khalifa (2020)*, *El-Alami, El Alaoui & Nahnahi (2022)*, *Mulki et al. (2019)*, *Albadi, Kurdi & Mishra (2019)*, *Husain (2020)*, *Muaad et al. (2022)*, *Aljarah et al. (2021)*, *Duwairi, Hayajneh & Quwaider (2021)*, *Anezi (2022)*, *Albadi, Kurdi & Mishra (2018)*, *Althobaiti (2022)*, *Almutairi & Al-Hagery (2021)*, *Al-Hassan & Al-Dossari (2022)*, *Shannaq et al. (2022)*, *Mouheb et al. (2019)*, *Aldjanabi et al. (2021)*, *Faris et al. (2020)* |
| 5 | AUC | *Alshalan & Al-Khalifa (2020)*, *Albadi, Kurdi & Mishra (2019, 2018)* |
| 6 | G-mean | *Aljarah et al. (2021)* |
| 7 | F-macro | *Alsafari, Sadaoui & Mouhoub (2020)* |
| 8 | Micro-average | *Althobaiti (2022)* |

techniques used for the classification process in the identified research on hate speech detection.

RQ4: What performance metrics are commonly used in HS identification for the Arabic Twitter dataset?

Measuring performance is a crucial aspect of text classification and machine learning research. Among the selected studies, eight evaluation metrics were identified, which are accuracy, precision, recall, F1-score, micro-averaging, AUC, geometric mean (G-mean), and F-macro. The analysis of the chosen studies reveals that, depending on the particular technological goals, several assessment metrics are used in distinct models for hate speech identification in Arabic tweets. Among the selected studies, the most widely used performance indicators were F1 score, precision, accuracy, and recall, with 22, 20, 17, and 20 studies respectively, as indicated in Table 11. Accuracy assesses the correct prediction of cases and is particularly applicable when all classes hold equal significance. It proves effective for evaluating unbalanced class distributions. In contrast, the F1 score is more suited when a balance between recall and precision is sought because it offers a stronger indicator of cases that were incorrectly classified. Contrarily, precision measures how accurately a model or system predicts the positive. Precision focuses on the percentage of positive instances that were accurately detected out of all positive instances that were projected to be positive. Precision is particularly crucial when performing jobs like fraud

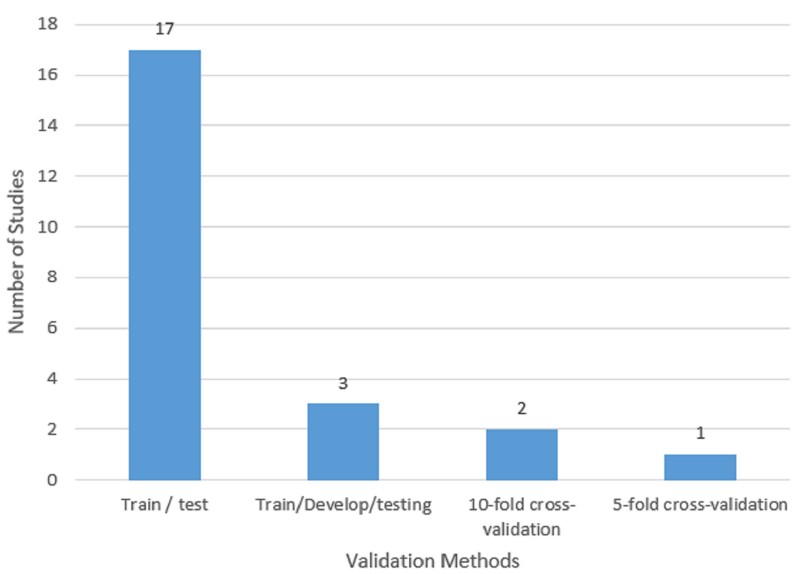

**Figure 8 Validation method used in the selected studies.**

identification or medical diagnosis where false positives can have costly or negative effects. A recall is a performance statistic frequently used in machine learning and binary classification applications. It is sometimes referred to as sensitivity or true positive rate. Recall quantifies the proportion of positive instances that are correctly identified by a model. Recall is concerned with how well a model can identify each positive case out of all the real positive instances in the data. Macro averaging determines the average, treating each class equally, after independently calculating the metric for each class. On the other hand, micro-averaging, which is favored when there are potential class imbalances, computes the average while taking into consideration the contributions of each class. The area under the curve (AUC) measures how well a classifier can distinguish between several classes. Higher AUC values show that the model is better at distinguishing between positive and negative classes. A performance indicator called the G-mean (Geometric mean) joins recall and precision to provide a comprehensive assessment of a model's efficacy, particularly in imbalanced classification problems. It takes into account both the ability to correctly classify positive instances and the ability to avoid misclassifying negative instances. The F-macro (macro-averaged F1 score) is a performance measure for evaluating the overall efficiency of a machine learning model in multi-class classification tasks. It calculates the average F1 score across all classes, giving equal weight to each class without considering class imbalance. While three studies employed AUC metrics to measure the effectiveness of hate speech identification in Arabic tweets, one study each employed G-mean, F-macro, and micro-averaging for model evaluation. The macro-averaged metric is suitable for highly unbalanced classes, While multi-classification problems are evaluated or represented using the AUC.

RQ5: What are the validation methods commonly employed in HS identification for the Arabic Twitter dataset?

A set of data that is not viewed during the model training phase is referred to as "validation" data. A model's performance can be evaluated more accurately by using unobserved data. This evaluation procedure is also referred to as a "train-test split algorithm." The training data is a set of information used to calibrate a machine learning model. An independent validation dataset is utilized to provide an objective assessment of how well the model matches the training data to fine-tune these parameters. The performance of the resulting model on the training set is then objectively evaluated using the test dataset. To understand the model, it is essential to underline the validation techniques used in this field of study. According to Fig. 8, three validation techniques were used in the chosen research. The first technique is called "k-fold cross-validation," and it involves breaking the primary training set into k smaller subsets. A method for assessing a machine learning model's performance is called k-fold cross-validation. The provided dataset is split into k subsets or folds that are roughly similar in size using this procedure. Then, each fold is utilized as the test set once while the remaining k-1 folds are used for training. The training and assessment process is then repeated k times. The model is trained on the training set (k-1 folds) and assessed on the test set (one fold) throughout each iteration. As a result, numerous performance measures can be calculated because each fold serves as a separate test set. After that, an average of the results from all k iterations is used to produce a more reliable estimate of the model's performance. K-fold cross-validation's key benefit is that it makes it possible to evaluate the model more accurately by using the whole dataset for both training and testing. It helps to assess how well the model generalizes to unseen data and provides a more accurate estimate of its performance. Common choices for the value of k are 5 and 10, but they can be adjusted based on the size of the dataset and the computational resources available. In the selected studies for this research, one of the studies applied five-fold cross-validation, and two studies employed ten-fold cross-validation The second method involves splitting the data into training, testing, and development sets. A further technique for assessing and improving machine learning models is the division of training, testing, and development sets. The available dataset is split into three distinct subsets using this method: the training set, the testing set, and the development set (also known as the validation set or holdout set). Utilizing the testing set, the trained model's effectiveness is evaluated. It has information that the model hasn't encountered before during training. To assess the model's accuracy, precision, recall, or other pertinent metrics, predictions on the testing set are reviewed. The results of this assessment are used to assess how effectively the model generalizes to fresh, untested data. To make the model more accurate, utilize the development set and optimize its hyperparameters. Hyperparameters are settings or configurations of the model that are not learned from the data but are manually set by the user. By evaluating the model's performance on the development set, different hyperparameter settings can be compared to select the one that yields the best result. In this research, three of the selected studies used training-testing-development sets.

The data are divided into training and testing sets using the third technique. To assess a model's performance in machine learning, the training and testing split is a frequent technique. The training set and the testing set are two distinct subsets that are created using

this method from the supplied dataset. By exposing the model to a labeled dataset, the training set is utilized to train the model. The model gains knowledge from this training data and modifies its internal parameters to reflect the relationships and patterns found in the data. The trained model's effectiveness is evaluated using the testing set. It includes information that the model hasn't seen before during training. The model's predictions or classifications are compared against the known labels in the testing set to evaluate its accuracy, precision, recall, or other relevant metrics. This evaluation provides an estimate of how well the model is expected to perform on new, unseen data.

The purpose of the training and testing split is to assess the model's ability to generalize. Testing the model on unseen data helps determine whether the model has learned meaningful patterns or if it has simply memorized the training data (overfitting). The training and testing sets should have a similar number of data points distributed throughout them and should be representative of the entire dataset. To produce an impartial and fair split, randomization is frequently used. As indicated in Fig. 8, a total of 17 studies utilized the training-testing sets, showing that it is the most used validation method in the selected studies.

## RESEARCH CHALLENGES AND FUTURE DIRECTIONS

This section provides the research challenges and potential research directions identified in the existing studies on speech detection. The visual representation of the challenges is depicted in Fig. 9 and the description of each challenge is provided below.

### Language and system barriers

The rapid evolution of language, especially among young people who often communicate on social media, requires ongoing research on hate speech datasets. For example, online platforms are taking steps to manually and automatically remove hate speech content (*Antoun, Baly & Hajj, 2020*; *Araci, 2019*). However, those who spread hate speech regularly strive to find new ways to avoid and bypass the restrictions imposed by these systems. For instance, some users post hate speech as images with text, which can bypass basic automatic hate speech identification. While converting image to text to regular text may address some specific issues, it presents several challenges Because of the constraints of this conversion process and the shortcomings of the automatic hate speech detection technologies currently in use. Additionally, changing the language structure poses another challenge, such as using unfamiliar abbreviations and mingling diverse languages. For example, (i) Creating sentences where a portion is written in Arabic while another portion is written in a different language., and (ii) phonetically writing sentences in another language (*e.g.*, writing Arabic sentences using English characters).

### Datasets issue

There is a lack of universally agreed-upon datasets considered optimal for automatically detecting hate speech. Authors tend to annotate datasets differently depending on their interpretation and the specific requirements of their tasks. Additionally, many datasets have been annotated through crowd-sourcing, which introduces concerns regarding the

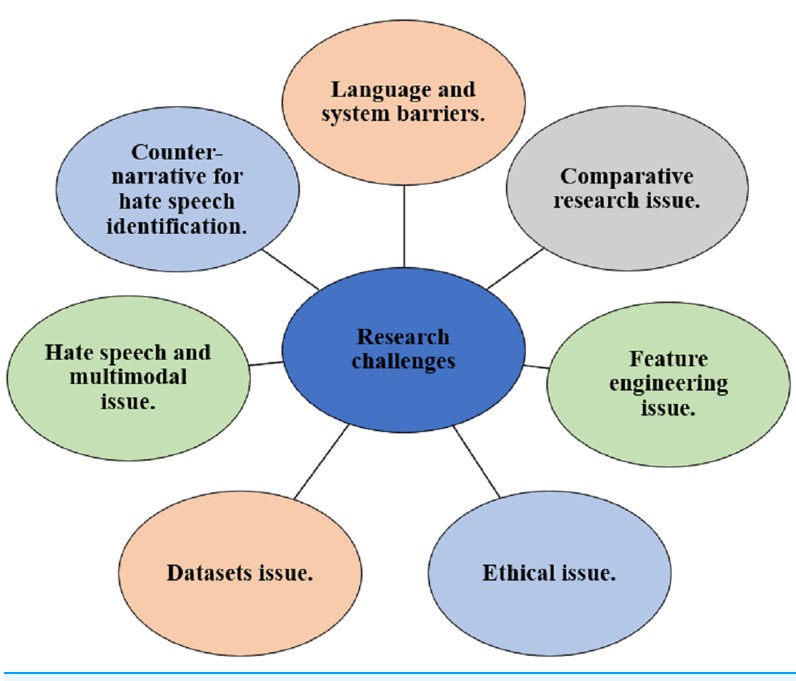

**Figure 9  Research challenges.**               

knowledge and expertise of the annotators. Dataset challenge discussions are grouped into three sub-categories, including clear label definition, annotation quality, and ethical issues, which are provided below.

### Clear label definitions

Defining clear labels is crucial for effectively recognizing hate speech as distinct from another objectionable language (*Davidson et al., 2017*). A dataset should encompass a wide range of hate speech categories, such as sexism, racism, abusive language, misogyny, and cyberbullying, to provide comprehensive coverage. This can be achieved through a multi-labeling approach, although some instances of ambiguity may arise, as observed in the labeling of racism and sexism (*Waseem, 2016*). Alternatively, a hierarchical method, as demonstrated in *Basile et al. (2019)* study on subtypes of hate speech and aggression, respectively, can also be employed.

Annotation quality: The characteristics of rude and hateful words pose challenges in establishing consistent annotation criteria, as it often leads to loose grammatical structures and cross-sentence boundaries (*Nobata et al., 2016*). Consequently, it is crucial to regularly update hate speech datasets to incorporate newly available knowledge. A study by *Poletto et al. (2021)* indicated that only around two-thirds of the datasets now in use report inter-annotator consensus, rules, definitions, and examples. To obtain high levels of inter-annotator agreement, it is necessary to provide extensive instructions and involve expert annotators. The quality of dataset annotation can significantly on whether annotators received thorough recommendations, bare-bones instructions, or no instructions at all. Another study conducted by *Yin & Zubiaga (2021)* on existing hate speech datasets found that of the 63 datasets, 43% failed to offer any rules for annotating, and 32% merely

provided hazy suggestions. Comparatively, a smaller selection of 25% (16 datasets) had thorough annotation guidelines. Recognizing the parameters of the underlying hate speech study is critically important, thanks to these rules. For instance, it is crucial to enumerate high-level characteristics like the number of employees, the maximum number of jobs per worker, the number of jobs completed by each employee, and the number of text annotations (*Sabou et al., 2014*). Environmental restrictions and user experience are other important aspects that can affect annotator involvement when utilizing an interface designed specifically for annotation. The accessibility of the source code is also helpful for the dissemination and generalization of research. According to *Sabou et al. (2014)*, with little effort being made to create artificial datasets, 98% of the datasets were gathered from social networks and manually tagged. or enriching existing ones.

### Ethical issue

Privacy poses a significant challenge in the extensive utilization of hate speech datasets. Such datasets are frequently gathered by leveraging content that genuine users, may not want to be identified as posted. Currently, the majority of researchers do not routinely get explicit agreement from all users whose content is examined Instead, they depend on implicit agreement because the content is either publicly available or semi-publicly available (*Williams, Burnap & Sloan, 2017*). Anonymization is frequently proposed as a primary solution to address this concern (*Fortuna et al., 2019*). Although this method is tried-and-true and reliable, it occasionally falls short of privacy standards because it is frequently easy to re-identify users using a variety of methods provided the dataset-gathering site is known. It is currently possible to recreate the full details of the posts *via* APIs, as many widely used datasets only offer annotations and post IDs. This is especially true for tweets, where posts become inaccessible whenever a person modifies their privacy settings or has their account suspended on Twitter. Additionally, some ethics experts contended that because the post ID immediately links to the user. However, this strategy does not entirely uphold ethical ideals, making it potentially worse than anonymization.

Additionally, it is important to consider dataset degradation, which introduces further challenges to the hate speech identification pipeline. For instance, the dataset released by *Waseem & Hovy (2016)* was gathered using Twitter's ID retrieval API, which caused significant degradation because the original tweets are no longer accessible. According to *Chung et al. (2019)*, within a year, the dataset by *Mathew et al. (2019)* nearly 60% of its content was lost. degrading the dataset has many dangers, including a higher chance of overfitting and a possibility of unbalanced classes because a lot of the hate material is removed when there are fewer data available. A different position is that datasets could be made accessible for training by imposing data-sharing agreements, as is the case in several businesses, mainly in the medical industry. An argument against this strategy is that people who publish hate speech on social networks frequently do so to convey their ideas, beliefs, and feelings, and they might not always be comfortable with having their content used to train an Ai system.

In conclusion, ethical artificial intelligence (AI) has advanced and gained renewed attention on the global agenda (*Bird et al., 2020*), resulting in privacy issues with the

pervasive usage of AI-based technology in society. This has led to discussions and the creation of regulations and legislation that directly impact the handling of online data and hate speech. Based on our knowledge, there have not been any effective efforts to share abusive language training datasets in the domain, except sporadic use of sites like GitHub or data sharing in scientific issues and contests.

## Feature engineering issue

Hate speech presents challenges when it comes to identification due to its lack of typical and distinctive characteristics, making it difficult to detect within a large dataset (*Zhang & Luo, 2019*). Deep learning serves as a valuable tool for extracting features, particularly in capturing the semantic aspects of hate speech tweets. Researchers have employed the integration of CNN and RNN to address this issue. Additionally, a study by *Alsafari, Sadaoui & Mouhoub (2020)* emphasized the effectiveness of choosing features as a possible tactic, enabling the selection of a limited set of highly predictive features. In this study, SVM were applied both with the initial feature set and without the improved feature set. Another approach presented by *Rasel et al. (2018)* involved dimension reduction techniques, such as LSA and SVD. To offer useful inputs for algorithms, these methods were integrated with widely used feature extraction methods, including bag-of-words, N-grams, and TF-IDF. However, previous studies overlooked text data, leading to the inclusion of word embeddings, sentiment analysis, and topical information through LSTM-CNN (*Al-Hassan & Al-Dossari, 2022*). While the feature selection procedure can improve the accuracy of models, various feature selection methods were analyzed based on detecting hate speech.

## Comparative research issue

Extensive comparative studies that genuinely contrast and compare different approaches to HS identification in Arabic Twitter data are lacking in the literature. This creates ample opportunities for future comprehensive comparative studies on hate speech, encompassing aspects such as preparing data, developing features, training models, and evaluating them. Furthermore, there is a scarcity of research focused on the labeling issue, which considers the targeted individual/group of hate speech posts and builds upon advancements in theoretical studies in social psychology and HCI. This flaw makes it difficult for automated hate speech identification algorithms to be used in practice and the ability to conduct meaningful comparison analysis. It also raises concerns about how such hate speech detection techniques would impact the experience of the correctness of the models in comparison to human moderation and actual users. Interdisciplinary research and organization partnerships are required to answer these problems. Additionally, it would be beneficial to compare various training methods, debiasing strategies, overfitting-prone models, and how dataset features affect performance. For instance, before designing the model, extracting features, and performing preprocessing, it would be beneficial to take the trade-off between domain-specificity, language patterns, and the motivation behind hate speech into account when undertaking transfer learning.

## Counter-narratives for hate speech identification

The core concept of counter-narratives involves directly intervening in discussions by providing textual defenses against hate speech content that are meant to mitigate its effects, thereby preventing its further spread (*Benesch, 2014*). Experts in education, psychology, linguistics, and NGOs have always provided counter-narratives, empowering users with logical arguments to withstand the effects of hate speech. An example of such efforts is the European Council's "We Can!" toolbox. Recently, the automatic creation of counter-narratives utilizing deep learning-based natural language generation technology has been investigated. These ideas, however, frequently don't contain enough high-quality data and tend to elicit sterile or repeated responses. According to what we know, CONAN (*Chung et al., 2019*) has presented a massive, multilingual dataset of well-produced hate speech/counter-narrative pairs, providing superior counter-narratives that are thought to be the best and most varied among other counter-narrative datasets (*Fanton et al., 2021*). Similarly, *Chung, Tekiroglu & Guerini (2021)* have suggested models for generating counterstories that emphasize educational and multilingual answers. They have developed a knowledge-driven pipeline that can produce appropriate and instructive English counter-narratives without experiencing hallucinatory phenomena. Their model training involved using GPT-2 and XNLG transformers (*Chi et al., 2020*). However, XNLG generations is deemed to be the most instructive by human annotators, whereas GPT-2 generations are deemed to be the most appropriate. A similar concept can be introduced to other languages, especially Arabic language. Likewise, another work by *Tekiroglu et al. (2022)* has proposed the use of models for learning languages, like BERT, T5 (*Raffel et al., 2020*), GPT/2, and DialoGPT (*Zhang et al., 2019*) for constructing arguments against hate speech. However, these counter-narrative creation algorithms have not yet been put to the fullest test in real-world situations.

## Hate speech and multimodal fusion

While there has been a rise in generating text-based hate speech data, the detection of hate speech content in multimedia data has received less scientific attention. Visual, auditory, and vocal modalities are typically used to identify hate speech and offensiveness in multimedia sources (*Rana & Jha, 2022*). This brings up at least two serious issues (*Yang et al., 2022*). The difference between the definitions of hate and sarcastic expression is the first issue. A difference in feature allocations caused by the domain gap is the second issue (*Tzeng et al., 2014*; *Yang et al., 2022*). This gap is brought about by various sources of sarcasm and hate datasets in image-text pairings that contain an infinite number of image types (such as posters, and plain text), together with tag-symbolized text. Direct knowledge transfer may not be successful due to the different feature disseminations.

To solve these issues, a multimodal model that has already been trained is tweaked specifically for feature learning (*Das, Wahi & Li, 2020*). Moreover, several researchers have investigated model fusion strategies (*Sandulescu, 2020*) and ways to enhance data (*Lee et al., 2021*; *Velioglu & Rose, 2020*), thereby enhancing the performance of hate speech identification. Existing studies in this field, however, mostly concentrate on building multimodal models without taking into account the impact of the uneven and widely

dispersed samples for diverse hate speech attacks. *Cai, Cai & Wan (2019)* published a multimodal dataset for sarcasm recognition that was created using image-text tweets and created a hierarchical fusion model as the foundation. Thus, several algorithms were developed for assessing sarcasm datasets as found in *Pan et al. (2020)* and *Wang et al. (2020)* study.

It is important to note that the BERT model, which uses Visual BERT, was created for multimodal training (*Li et al., 2019*). The latter is referred to as the "BERT of vision and language" because it was trained on both images and captions before Ensemble Learning. According to the study in *Velioglu & Rose (2020)*, their method placed third out of 3,173 competitors in the Hateful Memes Challenge with an accuracy score of 0.765 on the test set. As best as we can tell, just one research article (*Rana & Jha, 2022*) has proposed a multimodal deep learning system that integrates audio characteristics expressing emotion with semantic features to detect hateful content. The new Hate Speech Detection Video Dataset (HSDVD), which was compiled especially for multimodal learning, is also presented in this work.

Multilingual multimodal hate speech detection has not received nearly as much attention as English multimodal identification. An illustration of this kind of work is the research done by *Karim et al. (2022)* to adapt this idea to Bengali. Modern neural architectures, such as the Bi-LSTM/Conv-LSTM with word embeddings, the monolingual Bangla BERT, the multilingual BERT-cased/uncased, and the XLM-RoBERTa, were employed by the authors to simultaneously assess textual and visual data for hate speech identification. The best result, with an F1 score of 0.83, was produced by the combination of XML-RoBERTa and DenseNet-161. English multimodal detection using a comparable fusion model was successful (*Sai, Srivastava & Sharma, 2022*) and had a 67.7% accuracy rate. The same concept can be applied to the Arabic language to evaluate the effect on the predictive performance of different models.

## CONCLUSION

Recently, the extensive use of social media has increased some highly unwanted phenomena, including hate speech and occurrences connected to it. The impact of this research has a wide range of applications, including algorithm development, by informing the creation of a more accurate hate speech detection algorithm specifically tailored for Arabic tweets, policy rule regulation, by assisting decision-makers and legal authorities in understanding the scope and difficulties of hate speech on social media, community management, by assisting community managers in addressing hate speech and its impact on social media platforms, legal and ethical consideration, by highlighting legal and ethical issues that are relevant to the identification and management of hate speech, which can be crucial for legal professionals and ethicists working in this area. Despite the impact and the ongoing research attempts to address the issue of hate speech in Arabic tweets, there are still challenges in creating a workable answer to user-generated content. This research conducted a comprehensive review of existing literature, complementing previous reviews and surveys to promote this field's research. The selected studies provided insights into various aspects, including hate speech categories, types of the Arabic language, commonly

used classification techniques, performance indicators, and validation techniques employed. Additionally, a thorough examination of the chosen papers was performed to detect and outline the challenges and recommendations associated with identifying hate speech in the Arabic language. Future research is advised to handle the problems identified in prior studies, such as language and system barriers, dataset limitations, feature engineering challenges, comparative research gaps, counter-narratives for hate speech identification, and the fusion of hate speech and multimodal data. Moreover, exploring the issue of hate speech in languages that are not Arabic or on different social network platforms would be of great interest. This study consolidates the perspectives presented in published papers and serves as a valuable reference for researchers. It is crucial for further studies in which the research community can focus on developing advanced methods for hate speech detection tasks.

### Funding
This work was supported by the UM Research Maintenance Fee (RMF1510-2021). The funders had no role in study design, data collection and analysis, decision to publish, or preparation of the manuscript.

### Grant Disclosures
The following grant information was disclosed by the authors:
UM Research Maintenance Fee: RMF1510-2021.

### Competing Interests
The authors declare that they have no competing interests.

### Author Contributions
- Ali Alhazmi conceived and designed the experiments, performed the experiments, analyzed the data, performed the computation work, prepared figures and/or tables, and approved the final draft.
- Rohana Mahmud conceived and designed the experiments, performed the experiments, analyzed the data, authored or reviewed drafts of the article, supervision, and approved the final draft.
- Norisma Idris conceived and designed the experiments, performed the experiments, analyzed the data, authored or reviewed drafts of the article, supervision, and approved the final draft.
- Mohamed Elhag Mohamed Abo conceived and designed the experiments, performed the experiments, analyzed the data, authored or reviewed drafts of the article, supervision, and approved the final draft.
- Christopher Eke conceived and designed the experiments, performed the experiments, analyzed the data, performed the computation work, prepared figures and/or tables, and approved the final draft.

## Data Availability

This is a literature review; there is no raw data.

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
