# Peer review of "A systematic literature review of hate speech identification on Arabic Twitter data: research challenges and future directions"

_PeerJ Computer Science, doi:10.7717/peerj-cs.1966_

## Round 0.1 · original submission · Major Revisions

Thank you for submitting your manuscript to the PeerJ Journal. Please adjust all comments made by reviewers.

**Language Note:** The review process has identified that the English language must be improved. PeerJ can provide language editing services - please contact us at copyediting@peerj.com for pricing (be sure to provide your manuscript number and title). Alternatively, you should make your own arrangements to improve the language quality and provide details in your response letter. – PeerJ Staff

Reviewer 1 ·

Basic reporting

No comment

Experimental design

The methods are clearly described, and investigation meets the standards. However, although the authors initially found many papers they finally selected 24 studies. They should add some evidence that these small number of studies generate robust results for a scietific bibliometric analysis.

Validity of the findings

Authors should extend their comments on the results of analysis, as per example why they have found a large number of such studies for West Asia?

Reviewer 2 ·

Basic reporting

The aim of this study has been clearly stated as " The goal is to examine
25 the research trends in Arabic hate speech identification and offer guidance to researchers by
26 highlighting the most significant studies published between 2018 and 2023"

The application of this research should be presented wider and it should not be limited to the guidance for researchers

The way of presentation as for example Research method - in some part you could use graph or sth like that to make it more clear in presentation. Especially "Quality Assessment" is not transparent.

Most of the findings are described in pure technical way as how many sth etc and it is just written way of presentation of numbers on the figure or table as for example presented in Publication year overview. Numbers can be read from table/ figure while you should add some comment and possible meanings.

Apart from it it is quite wide and good systematic review.

Experimental design

The structure and a way of presentation should be more transparent . For example: in search strings - queries are hardly to separate from the rest of the text;

Validity of the findings

No comment.

Additional comments

No comment.

---

## Round 0.2 · Major Revisions

Thank you for your submission to PeerJ Computer Science.

It is my opinion as the Academic Editor for your article - A Systematic Literature Review of Hate Speech Identification on Arabic Twitter Data: Research Challenges and Future Directions - that it requires a number of Major Revisions accordance with review.

Please adjust your manuscript accordance with review comments and make changes in yellow colour.

Reviewer 1 ·

Basic reporting

The topic is well described and structured.

Experimental design

Methods are well described. Paper is in the scope of the journal. Review is well organized.

Validity of the findings

Originality is stated. Conclusions are supported by results. Authors added limitations and directions or further research

Additional comments

No further comments

Reviewer 2 ·

Basic reporting

The goal of this research has been described as " to examine the research trends in Arabic hate speech identification and offer guidance to researchers by highlighting the most significant studies published between 2018 and 2023".
Introduction part is too long. It is very easy to lose the plot. It should be shorten.
In the methodological part it is difficult to figure out that you use PRISMA methodology. You should mentioned about in the beginning. Also too many text in the methodology part and it would be better to present at one everything on the graph.
Research questions are formulated properly.
Results are interesting however I would suggest to think over about another form of presentation them - in more clear way - implement some table etc.

Experimental design

From the theoretical / logical point of view is ok, but it is not clear way of presentation.

Validity of the findings

Meet standards but modify the way of presentation .

Additional comments

As above.

---

## Round 0.3 · accepted · Accept

Respected Authors,

Thank you for your manuscript.

Reviewer 1 ·

Basic reporting

No comment

Experimental design

No comment

Validity of the findings

No comment

Additional comments

No comment

Reviewer 2 ·

Basic reporting

The quality of the paper has been improved , the comments have been taken into account.

Experimental design

Ok.

Validity of the findings

Ok.

Additional comments

No more comments.

·

Basic reporting

The paper is interesting. It has already been reviewed by reviewers and improved by authors. I have nothing to propose for improvements.
Figure 1 is a generic methodology for "how to develop articles". If the article were mine, I would not publish it, because it is too well known and does not bring anything new to the article.

Experimental design

The paper is interesting. I have nothing to propose for improvements

Validity of the findings

The conclusions are interesting

Additional comments

The paper is interesting and it can be published.